# Carbon emissions and radiative forcings from tundra wildfires in the Yukon-Kuskokwim River Delta, Alaska

Michael Moubarak[1], Seeta Sistla[2], Stefano Potter[3], Susan M. Natali[3,*], Brendan M. Rogers[3,*]

[1]Department of Biology, Hamilton College, Clinton, NY 13323, United States
[2]Department of Natural Resources and Environmental Sciences, California Polytechnic State University, San Luis Obispo, CA 93407, United States
[3]Woodwell Climate Research Center, Falmouth, MA 02540, United States

*Correspondence to:* Michael Moubarak (moubarak.michael@gmail.com)
*Equal last authorship

**Abstract**

Tundra environments are experiencing elevated levels of wildfire, and the frequency is expected to keep increasing due to rapid climate change in the Arctic. Tundra wildfires can release globally significant amounts of greenhouse gasses that influence the Earth's radiative balance. Here we develop a novel method for estimating carbon loss and the resulting radiative forcings of gaseous and aerosol emissions from 2015 tundra wildfires in the Yukon-Kuskokwim Delta (YKD), Alaska. We paired burn depth measurements using two vegetative reference points that survived the fire event —*Sphagnum fuscum* and *Dicranum spp.*— with measurements of local organic matter and soil carbon properties to estimate total ecosystem organic matter and carbon loss. We used remotely-sensed data on fire severity from Landsat 8 to scale our measured losses to the entire fire-affected area, with an estimated total loss of 2.04 Tg of organic matter and 0.91 Tg of carbon, and an average loss of 3.76 kg m$^{-2}$ of organic matter and 1.68 kg m$^{-2}$ of carbon in the 2015 YKD wildfires. To demonstrate the impact of these fires on Earth's radiation budget, we developed a simple but comprehensive framework to estimate the radiative forcing from Arctic wildfires. We synthesized existing research on the lifetime and radiative forcings of gaseous and aerosol emissions of $CO_2$, $N_2O$, $CH_4$, $O_3$ and its precursors, and fire aerosols. The model shows a net positive cumulative mean radiative forcing of 3.67 W m$^{-2}$ using RCP 4.5 and 3.37 W m$^{-2}$ using RCP 8.5 at 80 years post-fire, which was dominated by $CO_2$ emissions. Our results highlight the climate impact of tundra wildfires, which positively reinforce climate warming and increased fire frequency through the radiative forcings of their gaseous emissions.

**1 Introduction**

The Arctic region is characterized by permafrost soils with low rates of decomposition and high carbon content from millennia of positive net ecosystem production (NEP; Lindgren et al., 2018). As a result, there is more than twice as much carbon stored in permafrost soils as there is in the atmosphere, including roughly 1,035 ± 150 Pg of carbon in the top three meters of soil (Schuur et al., 2015). Surface air temperatures in the Arctic have been increasing more than twice as fast as the global average since the mid-20[th] century (Cohen et al., 2018) and are expected to continue increasing with more variable precipitation, hence more frequently pairing hot and dry conditions (IPCC, 2021; Hu et al., 2015). Changing environmental conditions will place some of this large carbon stock at risk for release into the atmosphere through increased biological activity and wildfire (Natali et al., 2019, Natali et al., 2015, Rocha et al., 2011a, Hu et al., 2010).

Fires are an increasingly important component of tundra carbon cycling.  Fires are becoming more frequent
in Arctic systems due to increasing occurrences of hot and dry conditions coupled with more lightning ignitions
(Chen et al., 2021, Bieniek et al., 2020, Veraverbeke et al., 2017b).  For example, the 2007 Anaktuvuk River
megafire on the North Slope of Alaska occurred during an especially hot and dry period and released an amount of
carbon similar in magnitude to annual sequestration across the entire tundra biome (Mack et al., 2011).  Continued
and potentially accelerated rates of warming are expected to further increase the frequency of tundra wildfires,
thereby releasing significant amounts of carbon and altering the net carbon balance of the tundra biome (Hu et al.,

69    2015).

A recent body of literature indicates the potential to estimate wildfire carbon emissions in boreal forests by
linking geospatial predictors, most prominently satellite-derived estimates of fire severity and extent, with *in situ*
measurements of carbon loss (e.g. Dieleman et al., 2020, Walker et al., 2018b, Veraverbeke et al., 2015b, and
Rogers et al., 2014). Despite increasing incidence of tundra wildfires, these studies have primarily focused on boreal
forests, and few estimates are available for tundra (Mack et al., 2011).  Whereas satellite-derived fire extent and
severity is widely available, representative *in situ* measurements of tundra wildfires are not. *In situ* measurements of
fire effects on tundra organic soils are challenging due to a lack of reference points (e.g., adventitious roots) that
survive the fire, which are used to estimate pre-fire organic matter depth and content (Walker et al., 2018a, Rogers et
al., 2014, Boby et al., 2010).  Measurements of organic matter loss must also be region-specific because of
differences in vegetation and soil properties (Walker et al., 2020c, Mack et al., 2011).
Although carbon loss estimates from Arctic wildfires are important for understanding the impacts of
climate change on carbon budgets, radiative forcings from wildfire gaseous and aerosol emissions are needed to
properly gauge the impact on the Earth's atmosphere and climate (Huang et al., 2016, Ward et al., 2012, Randerson
et al., 2006).  Radiative forcings from wildfires depend spatially on fire severity and atmospheric conditions and
temporally on changing atmospheric background concentrations in the months, years, and decades following the fire
(Huang et al., 2016, Joos et al., 2013).  To date, radiative forcings from high-latitude wildfires have been estimated
for particular boreal forests (Huang et al., 2016, O'Halloran et al., 2012, Randerson et al., 2006) and within global
models (Ward et al., 2012), but not for tundra ecosystems.
Here we take a two-step approach to assess the impact of tundra wildfires on carbon budgets and climate.
We first developed a method for measuring carbon loss *in situ* in tundra ecosystems, particularly for the 2015 fire
season in the Yukon-Kuskokwim Delta, AK. We tested the agreement of multiple pre-fire reference points for
estimating burn depth in the field and combined these estimates with laboratory-measured organic matter and carbon
fractions to estimate emissions. We then used 30m remotely-sensed fire severity data (differenced Normalized Burn
Ratio, or dNBR; Key and Benson, 2006) from Landsat 8 to scale our measurements to the entire fire area.  Finally,
we estimated the long-term radiative forcings of the fire season's gaseous and particulate emissions, including long-
lived greenhouse gasses (GHGs), ozone, ozone precursors, and aerosols, using a variety of published algorithms and
arctic-specific parameters when available.
**2 Materials and Methods**
**2.1 Study Area**
Field measurements were collected in the summer of 2019 in a burn scar from the 2015 fire season in the
Yukon-Kuskokwim Delta, AK (YKD; Fig. 1).  The burn scar we sampled was adjacent to an uninhabited but
regularly accessed Arctic field research outpost in the YKD established in 2016.  Base camp was situated at 61.2632
ºN, 163.2458 ºW, approximately 95 kilometers northwest of Bethel, AK, accessible by float plane and helicopter.
Sampling was done within about an 11-kilometer radius of the base camp (Fig. 1c).  Measurements from the field
were scaled to all fire scars in the YKD that burned in 2015 (Fig. 1b). We used fire perimeters from the Alaska
Large Fire Database (ALFD; Kasischke et al., 2002).

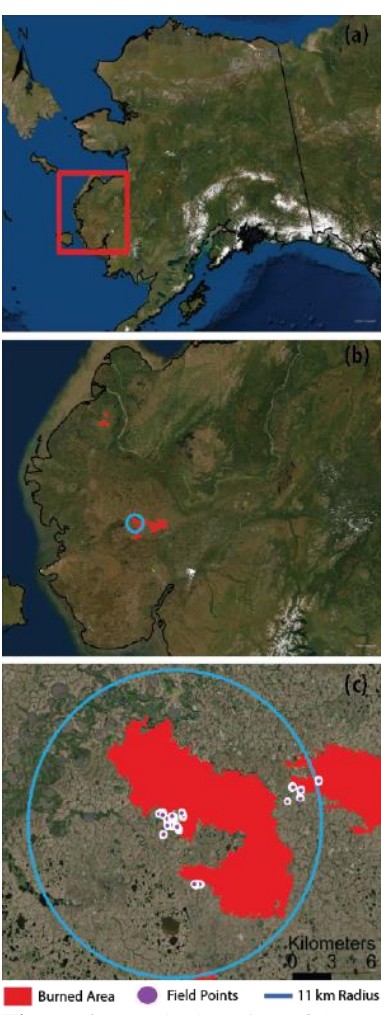

Burned Area ● Field Points ▬ 11 km Radius

**Figure 1.** (a) The location of the YKD within Alaska, (b) the areas of fires in 2015 within the YKD, and (c) the locations of our vegetative reference point height transects in burned and unburned areas within about an 11 km radius from base camp. Fire perimeters were derived from the Alaska Large Fire Database (Kasischke et al., 2002). Sources: Esri, DigitalGlobe, GeoEye, i-cubed, USDA FSA, USGS, AEX, Getmapping, Aerogrid, IGN, IGP, swisstopo, and the GIS User Community.

The YKD contains tussock-sedge, dwarf-shrub, and lichen and moss tundra communities over ice-rich

permafrost peat plateaus (Raynolds et al., 2005). These peat plateaus were the primary land component burned in

the wildfires and were separated by fens, bogs, and open water. Within our sample area, lichen was the dominant

vegetation type (Frost et al., 2020), shrubs were interspersed across the landscape, and tussocks were rare. Soil is

predominantly organic (Jorgenson et al., 2000), which has accumulated over sand deposits from the Pleistocene

period (Shaw, 1998).

**2.2 Fire severity**

Fire perimeters in the YKD in 2015 were extracted from the ALFD. We used a remotely sensed metric of

fire severity from Landsat 8 Tier 1 surface reflectance imagery: dNBR (Eq. 1), which is based on the Normalized

Burn Ratio (NBR; Eq. 2; Key and Benson, 2006). Clouds, cloud shadows, and snow were masked from all images
using the provided pixel quality attributes generated from the CFMASK algorithm (Foga et al., 2017). Imagery for
this index was acquired as the mean Landsat 8 composite between June 1$^{st}$ and August 31$^{st}$ for one year pre- and one
year post-fire, and for all fire scars. All remotely sensed variables were extracted in Google Earth Engine (Gorelick
et al., 2017). Where in Eq. (2) NIR is near-infrared (Band 5) and SWIR is shortwave infrared (Band 7).
$dNBR = (NBR_{prefire} - NBR_{postfire}) \times 1000,$         (1)
$NBR = \frac{(NIR-SWIR)}{(SWIR+NIR)},$         (2)
We compared dNBR from the YKD 2015 fire season to all fires in boreal and tundra Alaska to place the
measured dNBR in context. Using fire records from the ALFD, dNBR was acquired for all available fire perimeters
in Alaska. Differenced NBR was calculated in the same way as described before, but with the exception that
Landsat 4, 5, 7, and 8 were used in the pre and post-fire composites. Due to differences in spectral bands, Landsat 8
was corrected to match Landsat 7 using a regression technique (Roy et al., 2016). The first fire year we detected
dNBR was in 1989 and the most recent was in 2019. Boreal and tundra extents were defined using the United States
Environmental Protection Agency ecoregions, with tundra as level one class 2.0 and boreal as level one classes 3.0
and 6.0 (U.S. Environmental Protection Agency, 2010).
**2.3 Vegetative marker height measurements**
We measured the height of vegetative reference points above the surface in both unburned and burned
areas. Three main dominant surface vegetative reference points were available: *Sphagnum fuscum, Dicranum spp.,*
and *Eriophorum vaginatum* (tussocks; similar to the methods of Mack et al. (2011)). *S. fuscum* moss appeared in
large mats. *Dicranum* moss appeared in small dense patches. All vegetative reference points were particularly
conspicuous in burned areas as they were elevated above the burned surface (Fig. 2). In the burned area, we
measured the distance from the soil surface to top of the living parts of the vegetative reference points, which we
assumed to indicate full survival in the fire event. We did not measure dead remnants of vegetative reference points,
because we expect that these may not represent the actual pre-fire vegetation height.

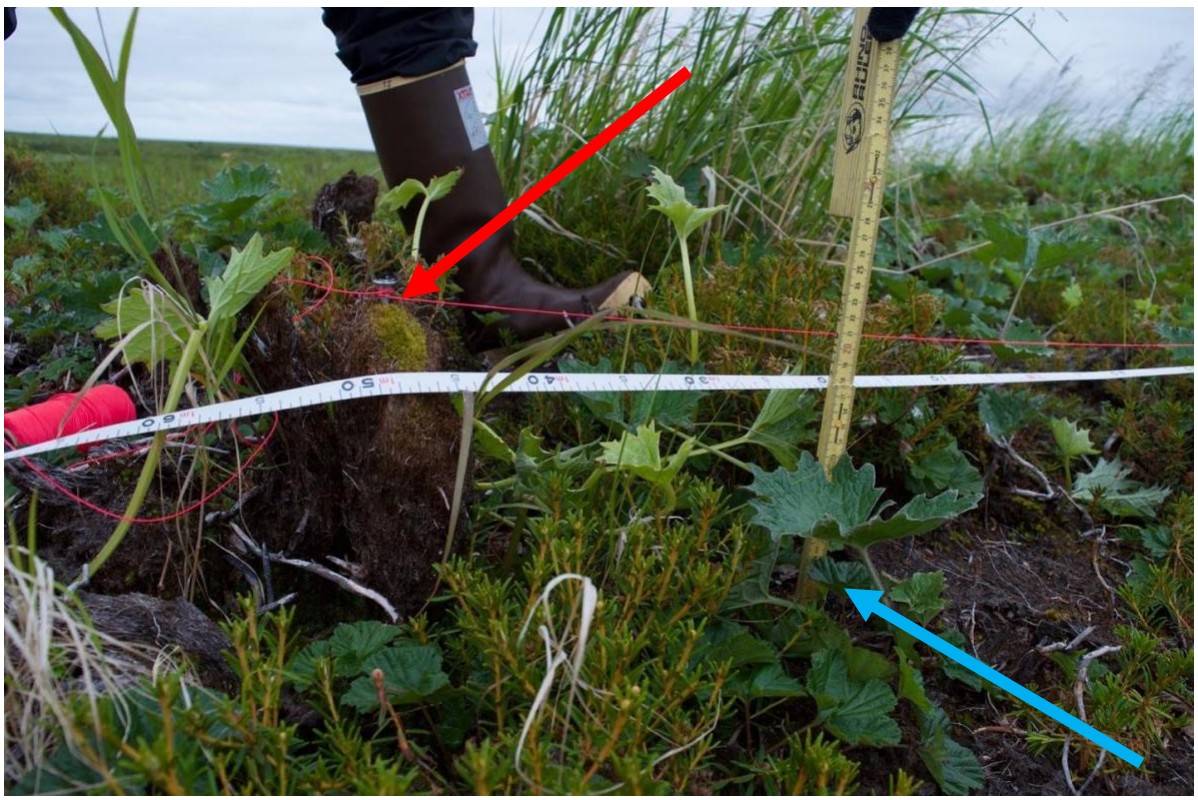

**Figure 2.** An example burn depth measurement from the surviving top of a *Dicranum spp.* moss patch (red arrow;
pink string) to the top of the soil (blue arrow) in a burned transect. Transect tape was used to measure distance
between the moss patches.
We measured the average height along a transect between two comparable reference points. At the highest

living point on each moss patch, or to the top of the corms at the base of a tussock, we inserted a nail (red arrow,
Fig. 2). We ran a taut piece of string between the two nails (Fig. A1). In between the edges of the moss patch or
tussock pair (Fig. A1), we took height measurements vertically from the string to the ground, which was soil in the
burned areas (blue arrow, Fig. 2) or a dense vegetative surface in the unburned areas (Fig. A2). In between the
starting and ending point measurements on the transect we recorded height every 25 centimeters (Figs. A2 and A3).
The starting and ending points were never greater than 50 centimeters from the nail. Latitude and longitude were
recorded at all transects with an accuracy of three meters (Fig. 1) for ground-truthing remotely sensed burn severity
categories.

One potential source of error is post-fire subsidence between the vegetative reference point pairs that may

bias the relative height of a vegetative marker above the surface (Jones et al., 2015). We minimized this potential
bias by maintaining transect lengths less than four meters, and predominantly one to two meters in burned areas.
We also visually assessed the area between each vegetative marker for signs of subsidence, including cracked soil or
large elevation differences, and chose not to sample these areas. Measuring from the top of one vegetative marker to
the top of another marker controlled for slopes and larger landscape elevation features because the string largely
mimicked the angle of the landscape.
We selected transects for height measurements opportunistically in the burned and unburned areas.
However, we tried to maximize spatial separation between transects and target visually identifiable areas of varying
burn severity. We corrected for spatial autocorrelation biases between transect averages in our statistical analyses
(see below). In unburned areas we maximized the number of transects measured for each vegetative reference point,
*Sphagnum* (n = 38), *Dicranum* (n = 40), and *Eriophorum* (n = 19).
Each burn depth measurement was associated with a dNBR value based on the 90 m pixel, the mean of
underlying 30 m pixels, containing its centroid. Burn depth measurements that came from the lowest third, middle
third, and upper third of dNBR values were considered to be in low severity/unburned, moderate severity, and high
severity areas, respectively, and a roughly even numbers of transects were sampled in each burn severity category
for *Sphagnum* (n = 56: high n = 20, mod n = 17, low/unburned n = 19), *Dicranum* (n = 54: high n = 18, mod n = 17,
low/unburned n = 19), and *Eriophorum* (n = 24: high n = 7, mod n = 11, low/unburned n = 6). Unburned patches of
tundra were sometimes included in low severity pixels within the burn scar, which is why these severity classes are
combined.
**2.4 Organic matter and carbon pool measurements**
We calculated the total ecosystem organic matter and carbon pool sizes in unburned areas surrounding the
fire scar. We took vegetation and soil samples together in cores using a hand drill and hollow metal drill bit that was
30 centimeters in length and six centimeters in diameter. Cores were extracted at three points (start, middle, and
end) along transects between like vegetative reference points. We selected four sites of less than half a hectare
surrounding areas where we took our unburned vegetative reference point height measurements. These sites were
selected opportunistically to ensure the presence of an appropriate number of each of the three vegetative reference
points. In each site, we extracted cores from three transects per vegetative reference point pair, which totaled 27 soil
cores per site. In one site we found only two tussock pairs, so our total number of samples was 105. We chose
unburned sites separated by at least one kilometer to control for heterogeneity in local soil and vegetation
characteristics.
We measured the height of the live vegetation layer and fibric soil horizons for each core.  Vegetation
layers and soil horizons were identified visually by soil density, texture, color, and identifiable plant parts.  Each
core was separated into vegetation and fibric soil layers that were homogenized, weighed, and subsampled in the
field, and subsequently stored frozen until analyzed.  In the lab, each vegetation sample and approximately 15 grams
of each fibric sample were dried for 48 hours at 60 ℃ and weighed to determine soil water content.  We used the
proportion of dry mass to wet mass to estimate the dry weight of the field sample, which was used to determine bulk
density (g dry mass $cm^{-3}$).  Soil samples were combusted for five hours at 450 ℃ to determine organic matter
content and analyzed for percent carbon using an Elementar Vario Max CN analyzer at the Woodwell Climate
Research Center, Falmouth, MA.  We assumed the dry mass of the vegetation layer was fully organic matter.  Using
the average across all herbaceous plant organs, we assumed the carbon content of the vegetation layer was 43.04%
from Ma et al. (2018).  To calculate organic matter and carbon pools (kg $m^{-2}$) for each organic soil sample, we
multiplied bulk density by the height of the layer and percent organic matter or percent carbon, respectively.  These
values were normalized for each sample to the average depth of the vegetation layer, which was seven centimeters,
and 10 centimeters in the fibric horizon.
**2.5 Calculating final estimates**
We calculated organic matter combustion and carbon loss across the fire area using burn depth derived
from the moss reference points. We excluded estimates derived from tussock measurements, because our sample
size of tussock-based measurements was substantially lower than the moss vegetative reference points due to their
infrequent occurrence in our study area (Frost et al., 2020).  Moreover, burn depth estimates from tussocks
correlated negatively with remotely sensed fire severity (Fig. A4).   This negative correlation with fire severity may
be attributable to mechanisms that are untestable with our sample size and study design, including altered burn
dynamics in close proximity to tussocks.  Future work could clarify the mechanism behind this discrepancy.
Excluding tussock measurements required reassigning dNBR values evenly as before across only *Dicranum* (n = 54:
high n = 18, mod n = 18, low/unburned n = 18) and *Sphagnum* (n = 56: high n = 20, mod n = 18, low/unburned n =
18) reference points.  The highest dNBR values in the low severity/unburned and moderate severity categories
became the threshold values for low severity/unburned to moderate severity and moderate severity to high severity
burn, respectively.  We split the fire area, based on these thresholds, into the three categories for burn severity.
There were 13,001 hectares of low severity burn, potentially including unburned patches, 8,516 hectares of moderate
severity burn, and 32,637 hectares of high severity burn, which summed to 54,154 hectares of total fire area (Fig.
A5).

For final analysis of organic matter and carbon loss, we averaged the height of *Sphagnum* and *Dicranum*

reference points in the unburned areas to determine one height for each reference point, since their respective
heights were statistically different ($p < 0.001$).  Burn depth was calculated at each transect in the burned areas by
subtracting the standard unburned height from the transect average height.  We also averaged the dry organic matter
and carbon pool within the vegetative layer and fibric horizon separately across *Sphagnum* and *Dicranum* reference
points to determine one pool size for our sampling area.  Average organic matter or carbon loss (kg m$^{-2}$) was
estimated as the product of the organic matter or carbon concentrations and depth of the vegetation and soil that
burned at each burn depth transect.  Average loss at each transect was the sum of loss from each horizon (see Fig.
A6 for schematic of organic matter and carbon loss calculation per transect). For our final calculation of total dry
organic matter and carbon loss, we averaged the organic matter and carbon lost across transects for *Sphagnum* and
*Dicranum* reference points within each fire severity category.  Over the landscape, total carbon or organic matter
released was calculated by multiplying average carbon or organic matter loss for a given burn severity category by
the total burned area for that category.  Total carbon or organic matter released was then calculated as the sum from
all three burn severity categories.
**2.6 Radiocarbon**

We used radiocarbon dating to confirm our burn depth measurements by comparing the radiocarbon age of

*Sphagnum* macrofossils on the burned soil surface with the radiocarbon age of *Sphagnum* macrofossils at different
depths of the unburned soil profile, similar to the methods of Mack et al. (2011).  In the field we extracted one
unburned core and three burned cores of the same dimensions as our primary soil cores, described above.  We
extracted a one-centimeter thick subsample of the cores every five centimeters in depth and froze the samples in a
sealed plastic bag until lab analysis.  To construct an age profile, we analyzed unburned cores at depths of five to 20
centimeters for radiocarbon age.  We assumed that a depth of zero centimeters had carbon from the time of harvest
in 2019. In the burned cores, we analyzed the zero-centimeter depth (i.e., surface) for radiocarbon.

In the lab, we visually identified *Sphagnum* branches in each of the subsamples being tested and rinsed

them with deionized water.  The branches were then dried for 48 hours at 60 ºC, and roughly five milligrams of dry
branch matter per sample was sent to the W.M. Keck Carbon Cycle Accelerator Mass Spectrometry Laboratory at
the University of California Irvine for the $^{14}$C content and $\delta^{13}$C. Results for $^{14}$C analysis are reported as the fraction
modern, which is defined as the ratio of the sample's carbon isotope ratio to that of a standard (Reimer, 2004).
Fraction modern values are calibrated to calendar years using OxCal online 4.4.1 (Ramsey, 2009), with the
appropriate modern era bomb curve taken from Hua et al. (2013). Due to the shape of the bomb curve, there are at
least two possible calendar years in which the carbon in the sample was fixed. However, we did not use the oldest
age for unburned samples with multiple calendar ages because the age of these samples must increase with depth
(Walker et al., 2019, Mack et al., 2011), which is not supported when considering the oldest calendar ages.
Furthermore, we were confident the burned samples were from the younger age because they were from visually
shallow burn depths. We compared the calendar age of carbon at the burned soil surface to age by depth profile
taken from the unburned core.
**2.7 Radiative forcings model**
We created a temporally-explicit model of radiative forcings for gaseous and aerosol emissions of tundra
wildfires and used it to compute the radiative forcings per unit burned area. The radiative forcings model was
driven using the average amount of organic matter lost across fire-wide burn severity classes and vegetative
reference points. Computing the radiative forcing of gaseous and aerosol emissions has been done for boreal fires
(e.g., Huang et al. 2016, O'Halloran et al., 2012, Randerson et al., 2006), but has yet to be applied to tundra systems.
Our model included the long-lived GHG species $CO_2$, $CH_4$, and $N_2O$, as well as short-lived climate forcers,
tropospheric $O_3$, $O_3$ precursors, and aerosols. Ozone precursors include $NO_x$, non-methanogenic volatile organic
carbons (NMVOCs), and CO.
We first used emissions factors from Akagi et al. (2011) to calculate the mass of gaseous and aerosol
emissions from our estimated organic matter losses. Emissions factors have not been previously defined for tundra
burning. Given that boreal forest, whose definition for emission factors includes organic soils, peat, and woody
vegetation, is likely the closest ecosystem type in terms of fuel properties to tundra in Akagi et al. (2011), we
employed these emission factors. However, we note a possible overestimate of relative contribution from woody
vegetation emissions in these numbers due to the relative lack of woody vegetation on the tundra landscape.
Furthermore, concentrations of gasses released from combustion of the same biomass type vary based on the
measurement technique, such as from differences in chemical mixing in laboratory, airborne, and ground-based field
methods (Akagi et al., 2011). Since Akagi et al. (2011) provides estimates of each emissions factor based on a
combination of values from previous studies, the emissions factors may capture uncertainty associated with the
variable methods of these source studies.

Once we estimated the mass of each gaseous emission, we calculated the concentration of the gas

remaining in the atmosphere and its radiative forcing each year after the fire season by synthesizing existing models
and research on the lifetimes and radiative forcings of these gaseous emissions.  Our calculations of radiative forcing
were dependent on the future ambient concentration of GHGs in the atmosphere.  As a result, we calculated the
radiative forcings of gaseous emissions for three scenarios: historic, representative concentration pathway (RCP)
4.5, and RCP 8.5.  The historic scenario assumes the ambient concentration of GHGs remains constant in the
atmosphere after the fire year.  Future atmospheric concentrations for each RCP were taken from Meinshausen et al.
(2011).  The radiative forcing for each gaseous emission per year was calculated separately and then summed across
forcing agent.  Since $O_3$ precursors and aerosols had the most uncertain lifetimes and radiative forcings (Bond et al.,
2011, Quinn et al., 2008), we calculated the total radiative forcing of the emissions with and without them.  A
flowchart of our general methodology for the entire radiative forcings calculation is presented in the appendix (Fig.
A7).
**2.8 Methane and Nitrous Oxide**

Our radiative forcing calculations for $CH_4$ and $N_2O$ were based on Ward et al. (2012).  For both gasses, we

assumed a pulse emission and calculated the concentration remaining in each year after the fires using a simple box
model with one outflow as shown in Eq. (3), where $C_o$ is the initial pulse concentration of the gas, t is the number of
years after the fire event, and L is the lifetime of the gas.  The initial pulse concentration of both gasses was
calculated by converting the mass emitted to a volume as a molar fraction of the atmosphere.
$C = C_o e^{\frac{-t}{L}},$                                                                                           (3)
$L = L_o \left(\frac{E}{E_t}\right)^{-0.05},$                                                                        (4)

We used the atmospheric lifetimes for both gasses reported in Myhre et al. (2013).  However, we calculated

a decreased lifetime of $N_2O$ using Eq. (4) from Ward et al. (2012), as its ambient concentration changed in future
emissions scenarios, where $L_o$ is the initial lifetime of $N_2O$ from Myhre et al. (2013), E is the ambient concentration
of $N_2O$ in the year the fires burned, and $E_t$ is the ambient concentration of $N_2O$ at each year after the fire event based
on the future emissions scenario (Meinshausen et al., 2011).  We did not account for the effect of the changing
atmospheric concentration on the lifetime of $N_2O$ due to the pulse emission itself.  We then calculated the
perturbation concentrations of $CH_4$ and $N_2O$ in the atmosphere each year after the fire event as the sum of the
remaining pulse emissions and the ambient concentrations and used Eqs. (5) and (6) from Ward et al. (2012) to
estimate the radiative forcing of both gasses. The radiative forcing of the gas at each year was RF, M was the
perturbation concentration of the gas whose radiative forcing was being calculated, and $M_o$ and $N_o$ were the ambient
concentrations of both gasses depending on which gas's radiative forcing was being calculated.
$$RF = 0.036(\sqrt{M} - \sqrt{M_o}) - [f(M, N_o) - f(M_o, N_o)], \tag{5}$$
$$f(M, N) = 0.47 ln[1 + 2.01 \times 10^{-5}(MN)^{0.75} + 5.31 \times 10^{-15}M(MN)^{1.52}], \tag{6}$$
**2.9 Carbon Dioxide**
Carbon dioxide has a highly variable lifetime under different future emissions scenarios depending on the
strength of ocean and land sinks. To account for this variation, we used impulse response functions (IRFs) from
Joos et al. (2013), which represent the fraction of a pulse of $CO_2$ remaining in the atmosphere at each year after the
pulse for each scenario. For each year post-fire, we multiplied the relevant IRF function by the concentration of
$CO_2$ initially released to estimate the amount of $CO_2$ remaining in the atmosphere. Similar to $N_2O$ and $CH_4$, the
initial concentration of $CO_2$ was calculated by converting the mass emitted to a volume as a molar fraction of the
atmosphere. The radiative forcing for $CO_2$ was calculated using Eq. (7) taken from Myhre et al. (1998), where C is
the perturbation concentration and $C_o$ is the ambient concentration of the gas in each year.
$$RF = 5.35 \ln\left(\frac{C_o + C}{C_o}\right), \tag{7}$$
**2.10 Tropospheric Ozone**
Tropospheric $O_3$ is created photochemically in the smoke plumes of wildfires by the combination of
sunlight and $O_3$ precursor gasses, and its creation can be expressed as a function of time and the amount of carbon
monoxide released (Jaffe and Wigder, 2012). We used data compiled in Jaffe and Widger (2012), who provided a
synthesis of $O_3$ and its relationship with CO for fire plumes sampled in the Arctic at various times after a fire to
calculate a mean plume lifetime and derive a linear relation between plume age and the ratio of $O_3$ to CO. Because
the average plume age sampled was five days, we assumed $O_3$ was created for five days after the fires. The
concentration of $O_3$ on each of those five days was calculated by multiplying the initial concentration of CO released
from the fires, calculated by converting the mass emitted to a volume as a molar fraction of the atmosphere, by the
ratio of $O_3$ to CO we estimated for that day. After five days, we assumed no more $O_3$ was created, and we
calculated the remaining amount of $O_3$ after day five using the box model approach from Eq. (3). The concentration
of $O_3$ on day five was $C_o$, and L was the lifetime of $O_3$ taken from Myhre et al. (2013). We converted the
concentration of $O_3$ to Dobson units (DUs) and converted from DUs to radiative forcing using a conversion factor
from Myhre et al. (2013) each day after the fire event.  The radiative forcing in each year was the mean radiative
forcing of all the days within that year, although the annual forcing was negligible after the first year.
**2.11 Ozone Precursors and Aerosols**
We used a method based on global warming potential (GWP), similar to Huang et al. (2016), to calculate
the radiative forcing of the ozone precursors and the direct aerosol effect.  GWP is defined as the ratio of the time-
integrated radiative forcing of a pulse emission of a gaseous species to the time-integrated radiative forcing of an
equivalent emission mass of $CO_2$ (Joos et al., 2013).  GWP is typically calculated for 20- and 100-year time
horizons.  Here, we use GWPs to determine the warming effect of the precursor and aerosol emissions relative to
$CH_4$ at the 20- and 100-year time horizons, as shown in Eq. (8).  We derived the radiative forcings for ozone
precursors based on $CH_4$ because the radiative forcing of ozone precursors is through their effect on methane over
the long-term (Collins et al., 2013).  To remain consistent with our methodology for ozone precursors, we derive the
radiative forcings for aerosols from $CH_4$ as well.  Furthermore, the radiative forcing of $CH_4$ has the simplest
derivation in our model, so we assume it is the strongest continuous radiative forcing from which to build our
continuous model using GWPs.
$$R_t = \frac{GWP_{t,x} \times EF_x}{GWP_{t,CH_4} \times EF_{CH_4}},$$ (8)
The emissions factor (EF) and GWP are defined for gaseous species x, and the GWP is defined at the time
horizon t.  For CO and NMVOCs, we interpolate this relative warming effect ($R_t$) between year one and year 20 by
assuming it remains constant.  To interpolate this effect between years 20 and 100, we mirror the shape of methane's
cumulative radiative forcing curve over that time interval.  From this curve we calculated the difference between
methane's cumulative radiative forcing at years 20 and 100.  We then calculated the fractional decrease of this
difference each year between years 20 and 100.  We multiplied those fractional decreases by the difference in $R_t$ at
years 20 and 100 for CO and NMVOCs to derive a relative warming effect for the precursors.  Finally, we
multiplied $R_t$ by methane's cumulative radiative forcing curve to estimate the cumulative radiative forcing for each
precursor every year after the fire event.  We used the global GWPs for CO and NMVOCs from Myhre et al. (2013).
Since $NO_x$ has a positive GWP at the 20-year time horizon and a negative GWP at the 100-year time
horizon, we chose not to mimic the shape of methane's radiative forcing curve when interpolating between the two
time points.  Instead, we calculate an $R_t$ value for $NO_x$ at year one by scaling $R_{20}$ in proportion to the change of
methane's cumulative radiative forcing from years one to 20. Then we interpolated between the $R_t$ values in years
one, 20, and 100 using an exponential decay function of the form $ae^{-bx} + C$, whose coefficients were calculated
using the "nls" function in R (R Core Team 2020). We multiplied the $R_t$ value for $NO_x$ at each year by the
cumulative radiative forcing of $CH_4$ in that year to obtain a cumulative radiative forcing curve for $NO_x$. We use the
global GWP for $NO_x$ from Myhre et al. (2013).

The radiative effect of aerosol emissions happens within a year of the fire event, as fire aerosols are

typically removed from the atmosphere via wet and dry deposition within a matter of weeks (Bond et al., 2011,
Quinn et al., 2008). We assumed that the cumulative radiative forcing of aerosols at any year after the fire event
would be constant and equal to the radiative forcing of aerosols in the year of the fire event. Therefore, to calculate
the cumulative radiative forcing from black and organic carbon direct effects in every year, we used $R_{20}$ and
multiplied this value by the cumulative radiative forcing curve of $CH_4$ at year 20. Radiative forcings of black and
organic carbon were summed to report a single value for aerosols. We used the GWPs for black and organic carbon
estimated for open biomass burning including the cryosphere effect from Bond et al. (2011). To estimate the
indirect aerosol effect, we multiplied the radiative forcing of the direct effect of aerosols each year by the ratio of
indirect to all sky direct effect radiative forcing from wildfires defined in Ward et al. (2012).
**2.12 Statistical analyses**

All statistical analyses were completed in R (R Core Team 2020) using the nlme package (Pinheiro et al.,

2020). For the vegetative reference point heights, we used the nlme package function "gls" to fit a linear model
using the generalized least squares method with average burn depth and height above the dense vegetation layer
along the transect as the response variable in the burned and unburned areas respectively. Within the "gls" function,
both models were corrected for spatial autocorrelation between transect locations by choosing the model with the
lowest Akaike information criterion (AIC) score across five correlation structures, exponential, Gaussian, linear,
rational quadratic, and spherical, to be our final model. In burned areas, the model had vegetative reference point
nested within burn severity type, while in unburned areas, we only modeled the effect of vegetative reference point
type. The model with the lowest AIC score for both burned and unburned areas had a rational quadratic correlation
structure. To analyze differences in organic matter and carbon pools, height, and bulk density of the vegetation and
fibric soil layers we used the "lme" function, defined in the nlme package, to fit linear mixed effect models with the
restricted maximum likelihood method. These models had the soil or vegetation layer characteristic value for each
core as their response variable, vegetative reference point type as a fixed effect, and transect number nested in site as
random effects.  Finally, we created linear fit models using the "gls" function with organic matter combusted and
carbon lost at each transect in the burned areas as response variables with the same structure as the burn depth
model.  These models had the lowest AIC scores with a rational quadratic correlation structure.  An analysis of
variance (ANOVA) was used to test for significant differences between groups for all of our models.
**3 Results**
We found that burn depth measurements were consistent with burn severity classification, and burn depth
increased with fire severity (Fig. 3).  Based on a two-factor ANOVA with vegetative reference point nested within
burn severity, differences in average transect burn depth were significant ($p < 0.0001$) between burn severity
categories.  The ranges of moderate severity burn depth measurements overlapped substantially with the high
severity and low severity/unburned measurements.  Moderate severity groups also had the most evenly distributed
probability densities across burn depth.  *Dicranum* measurements captured the shallowest and the deepest burn
depths (Fig. 3).  Nevertheless, across both vegetative reference points, there was a clear stepwise increase in burn
depth across burn severity classifications.

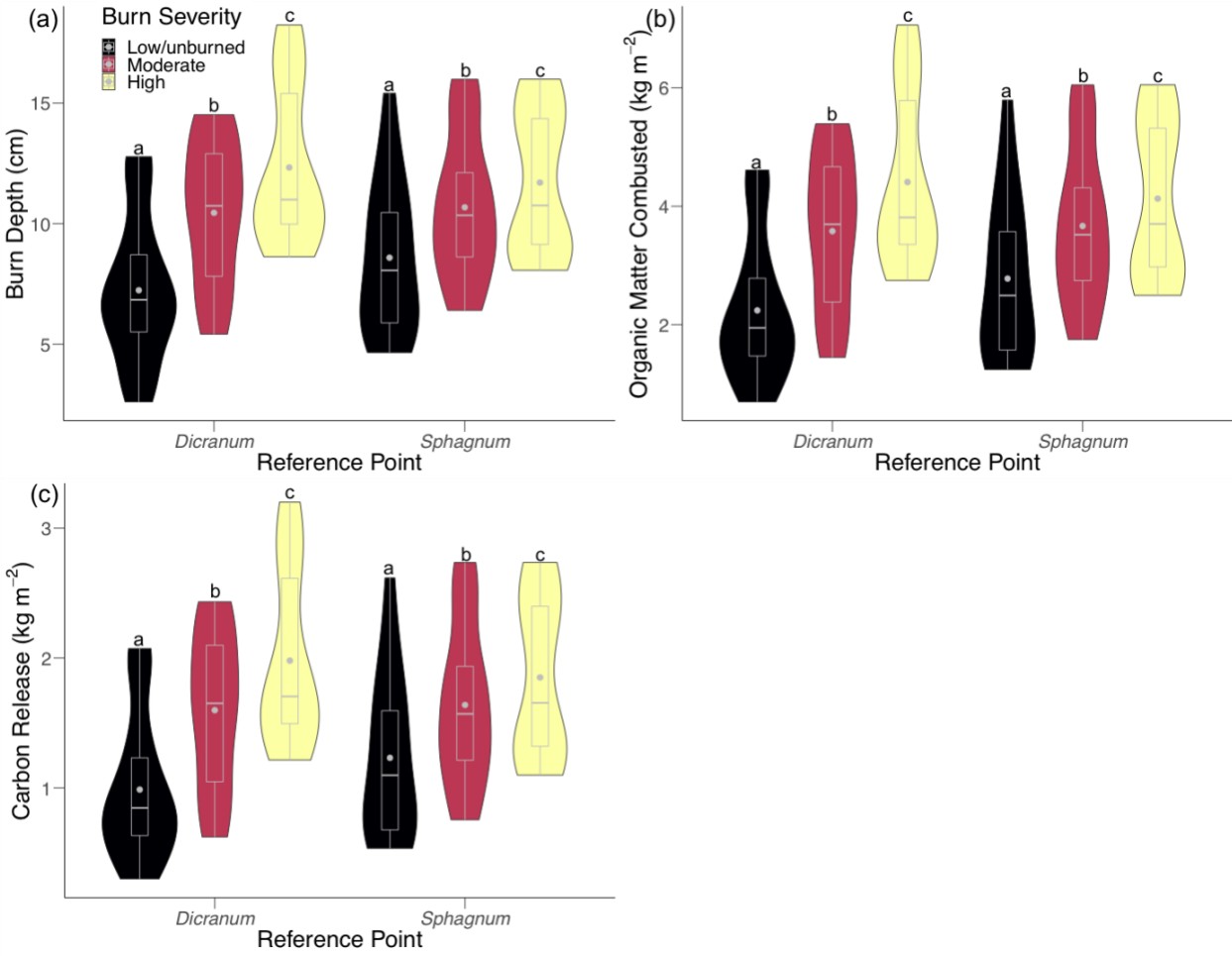

**Figure 3.** Distributions of average (a) burn depth, (b) organic matter combusted and (c) carbon released per unit burned area per transect within each reference point across burn severity levels. Boxes encompass the middle 50% of data, whiskers are the upper and lower quartiles, horizontal lines intersecting boxes show the median, and gray points are the mean. Letters indicate significantly different groups of data.

We found no significant differences between vegetative reference point types in terms of mean organic matter pool size, carbon pool size, height, or bulk density for either the live vegetation layer or fibric soil horizon extracted in our unburned cores ($0.3504 < p < 0.9558$). In the unburned areas, the fibric soil horizon ($17.9 \pm 0.63$ cm) was nearly three times thicker than the vegetation layer ($6.19 \pm 0.60$ cm) to the depth that our cores reached (~30 cm; Table 1), which is likely an underestimate of the soil organic layer depth, given that our cores only occasionally reached the mineral soil layer. However, there was larger variation in the vegetation layer ($1\sigma = 6.9$ cm) than fibric horizon ($1\sigma = 2.6$ cm) height. In most cases, the fire burned through the vegetation layer, but never completely through the fibric horizon (Fig. 3, Table 1). The fibric layer (to 24.1 cm average) had almost three times

greater organic matter and carbon pool sizes compared to the vegetation layer because of its higher bulk density and
height (Table 1).
**Table 1.** Bulk density, height, and carbon and organic matter percent and pool sizes of vegetation and fibric layers
measured in unburned soils averaged across reference points. Carbon and organic matter pool measurements are the
product of bulk density and their percent contents and are normalized to the height of the vegetation layer and 10
centimeters in the fibric layer. Sample size is 72, which is equally attributable to the two reference points. Error is
reported as the standard error of the mean (SEM).

| Layer | Bulk Density (kg m$^{-3}$) | Height (cm) | Percent Carbon (%) | Carbon Pool (kg m$^{-2}$) | Percent Organic Matter (%) | Organic Matter Pool (kg m$^{-2}$) |
|---|---|---|---|---|---|---|
| Vegetation | 26.7 ± 1.4 | 6.19 ± 0.60 | 43.04* | 0.712 ± 0.037 | 100* | 1.65 ± 0.09 |
| Fibric | 54.6 ± 5.0 | 17.9 ± 0.63 | 39.41 ± 0.56 | 2.07 ± 0.16 | 88.12 ± 1.17 | 4.48 ± 0.30 |

*Based on assumption
Carbon and dry organic matter loss estimates similarly increased with burn severity (Fig. 3, Table 2).
Based on a two-factor ANOVA with vegetative reference point nested in burn severity, average organic matter and
carbon loss differed between burn severity categories (carbon loss: $p < 0.0001$; organic matter loss: $p < 0.0001$) (Fig.
3). Similar to burn depth distributions, the moderate severity range overlapped with the high severity and low
severity/unburned distributions (Fig. 3).
**Table 2.** Carbon and organic matter loss averaged within each burn severity category across reference point, and
total carbon and organic mass loss over each burn severity category area. Total loss per burn severity category is
summed to calculate total loss over the total fire area by burn severity classification. Error is reported as SEM,
except for total loss over the fire area which is reported as the sum of errors.

| Severity | Sample Size | Area (hectares) | Carbon Loss | | Dry Organic Matter Loss | |
|---|---|---|---|---|---|---|
| | | | Average (kg m$^{-2}$) | Total (Tg) | Average (kg m$^{-2}$) | Total (Tg) |
| Low/unburned | 36 | 13,001 | 1.11 ± 0.10 | 0.144 ± 0.013 | 2.51 ± 0.22 | 0.326 ± 0.029 |
| Moderate | 36 | 8,516 | 1.62 ± 0.10 | 0.138 ± 0.009 | 3.62 ± 0.22 | 0.309 ± 0.019 |
| High | 38 | 32,637 | 1.93 ± 0.12 | 0.629 ± 0.039 | 4.29 ± 0.23 | 1.40 ± 0.08 |
| Total | 158 | 54,154 | | 0.911 ± 0.039 (Tg) | | 2.04 ± 0.09 (Tg) |

The 2015 fires burned a total of 54,154 hectares in the YKD. The distribution of dNBR values for our
focal fires in the YKD was intermediate compared to dNBR values from all tundra fires and on the low side but
within the range of dNBR values for all boreal fires in Alaska between 1989 and 2019 (Fig. A8). Roughly 60% of
the burned area was classified as high severity, while moderate severity and low severity/unburned accounted for
about 16% and 24% of the burned area, respectively (Table 2). As a result, total organic matter and carbon loss
within the high severity burn classification was over three times greater than the other two burn severity categories.
Although moderate severity occupied less fire area than the low severity/unburned, it contained roughly equal total
organic matter and carbon loss to the low severity/unburned areas due to about 50% higher organic matter and
carbon losses per unit area on average.  However, it should be noted that losses from low severity/unburned areas
are likely overestimates, given the inclusion of unburned areas within regions of low severity burn.  Summing
carbon loss and dry organic matter loss over the area of each burn severity category yielded a total loss of 2.04 ±
0.09 Tg of dry organic matter and 0.911 ± 0.039 Tg of carbon.  This magnitude of total loss corresponded to an
average 3.76 kg m$^{-2}$ of organic matter and 1.68 kg m$^{-2}$ of carbon loss across the fire area.

The age of carbon increased by ~50 years from 5 to 20 cm in depth in the unburned area.  Surface samples

from burned soils in a low severity/unburned area were dated to 2009, indicating a burn depth of about 5 cm or less,
which is within the range of our low severity/unburned burn depth measurements (Table 3).  The rate of change in
carbon age between 0 and 5 cm was 2.4 yr cm$^{-1}$; for the 15 to 20 cm interval, it was 3.6 yr cm$^{-1}$.  Because none of
our burn depth measurements exceeded 20 cm, all carbon released from the fires was likely relatively new carbon,
younger than 64 years of age.  Our average burn depth of 10.3 cm across burn severity and reference point indicated
an average age of about 23 years for carbon released.
**Table 3.** The fraction of the modern standard and calibrated calendar age for each radiocarbon sample.  Calendar
ages are the intercepts for the fraction modern value with the calibration curve.  We do not consider the oldest
calendar ages.

| Burn Status | Depth (cm) | Fraction Modern | Calendar Ages CE |
|---|---|---|---|
| Burned | 0 | 1.0486 ± 0.0020 | 1957, 2009 |
| Burned | 0 | 1.0493 ± 0.0024 | 1957, 2009 |
| Burned | 0 | 1.0403 ± 0.0018 | 1957, 2008, 2009 |
| Unburned | 5 | 1.0625 ± 0.0020 | 1957, 2007, 2008, 2009 |
| Unburned | 10 | 1.1198 ± 0.0023 | 1958, 1996 |
| Unburned | 15 | 1.4603 ± 0.0024 | 1963, 1974 |
| Unburned | 20 | 1.0124 ± 0.0017 | 1956 |

The radiative forcing for gaseous and aerosol emissions from the 2015 fire season was positive for 80 years

post-fire according to our model, regardless of the future atmospheric gas concentration scenario (Fig. 4a).
However, the magnitude of the fire emissions' radiative forcings decreased with increasing ambient atmospheric gas
concentrations, with RCP 8.5 having the lowest radiative forcing at 80 years post-fire (Fig. 4a).  Given the lack of
qualitative difference across future scenarios, below we present results for the RCP 4.5 scenario only (Table 4, Fig.
4b).  Nitrogen oxides were the only gas with a negative radiative forcing at the 80-year cumulative mean.  However,
they had a positive radiative forcing at the 20-year time horizon (Table 4).  The only other negative radiative forcing
came from the indirect effect of aerosols that completely counterbalanced the positive radiative forcing from other
gaseous emissions in the first year, but the magnitude of this effect in terms of cumulative integrated forcing
diminished rapidly after the first decade (Fig. 4b).  $CO_2$ produced the highest radiative forcing through 80 years
(Table 4, Fig. 4b). Nitrous oxide sustained its positive radiative forcing; however, the radiative forcing was of
intermediate magnitude with respect to all gaseous emissions. In contrast, ozone and carbon monoxide exhibited
high radiative forcings, but the magnitude of these diminished rapidly with respect to carbon dioxide. Likewise, the
effect of the other ozone precursors, aerosols, and methane decreased relatively rapidly (Table 4). Overall, 20-years
post-fire, the effect of ozone precursors increased the total radiative forcing by roughly 11%. However, this percent
increase drops to 5% after 80 years, and the ozone precursors and aerosols show little effect on the percent reduction
of the total radiative forcing between 20- and 80-years post fire. Therefore, the positive radiative forcing is
sustained by long-lived gaseous emissions, especially carbon dioxide.

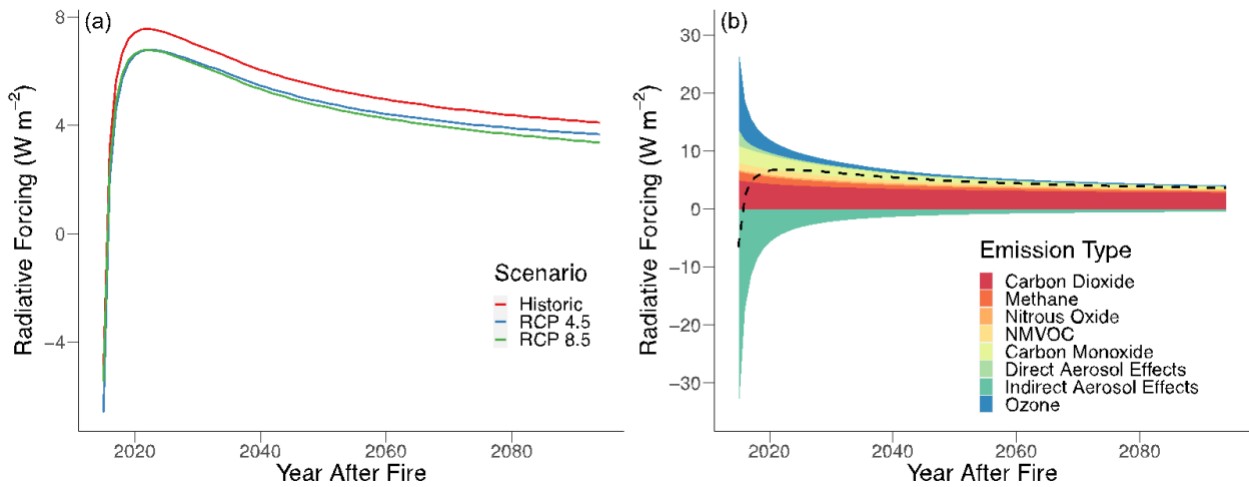

**Figure 4.** (a) Total cumulative mean radiative forcing of gaseous emissions for three future atmospheric gas
concentration scenarios, and (b) cumulative mean radiative forcing of different emissions (dashed line is net forcing)
under the RCP 4.5 future scenario over 80 years post-fire from the YKD 2015 fire season.
**Table 4.** Cumulative mean radiative forcing at 20- and 80- year time horizons post-fire for different gaseous and
aerosol emissions from the 2015 fire season under the RCP 4.5 future scenario. Percent reduction refers to the loss in
radiative forcing when considering an 80-year time horizon compared to 20 years.

| Time Horizon | Radiative Forcing (W m$^{-2}$) | | | | | | | | | | |
|---|---|---|---|---|---|---|---|---|---|---|---|
| | $CO_2$ | $CH_4$ | $N_2O$ | $O_3$ | NMVOC | CO | $NO_x$ | Indirect Aerosol Effects | Direct Aerosol Effects | Total with Ozone Precursors and Aerosols | Total without Ozone Precursors and Aerosols |
| 20 years | 3.69 | 0.76 | 0.28 | 0.67 | 0.62 | 1.51 | 0.03 | -1.71 | 0.14 | 5.99 | 5.41 |
| 80 years | 2.87 | 0.24 | 0.22 | 0.17 | 0.19 | 0.40 | -0.01 | -0.43 | 0.03 | 3.67 | 3.49 |
| Percent Reduction (%) | 22 | 68 | 21 | 75 | 69 | 74 | 133 | 75 | 79 | 39 | 35 |

**4 Discussion**
Here we describe a tundra wildfire season that combusted significant amounts of carbon and had a positive
warming effect on the atmosphere due to its gaseous and aerosol emissions over an 80-year time horizon. Our
findings suggest that increasingly frequent tundra fire regimes are an under-studied source of global GHG
emissions.  The 2015 fire season in the YKD represents a large efflux of carbon from a wetland tundra ecosystem,
generally not considered to be highly vulnerable to fires.  In total, we estimate that about 0.911 Tg of carbon was
released from 54,154 hectares of burned area in the YKD.  This fire area is roughly three times the 17,000 hectares
of tundra area burned on average annually between the years 2001 and 2018 in Alaska, but it only accounts for a
little more than half of the 83,000 hectares of Alaskan tundra that burned in the year 2015 (Scholten et al., 2021).
Over the same time frame, the 2015 fire season was the largest to burn in the YKD, with the majority of years
burning less than 10,000 hectares (Scholten et al., 2021).  Total carbon loss was driven by a similar per unit area
carbon emission rate as the 2007 Anaktuvuk River fire (Mack et al., 2011), which was roughly twice the size of the
fire season reported here and burned predominately moderate to high severity in a tussock tundra ecosystem (Jones
et al., 2009).  While the current status of tundra ecosystems as a sink or source of carbon remains uncertain, the 2015
YKD fire season alone released about a tenth of the carbon sequestered annually from tundra sink estimates
(Virkkala et al., 2021).  If tundra ecosystems become a source of carbon to the atmosphere under warming,
emissions from tundra wildfires may further exacerbate this positive feedback to warming due to a changing balance
between respiration and productivity (Natali et al., 2019, Belshe et al., 2013).  Note that we did not account for post-
fire effects on net ecosystem exchange of $CO_2$, which tend to be sources to the atmosphere with greater fire severity
due to continued soil respiration until sufficient vegetation recovers (Rocha and Shaver 2011a).

The tundra fires in the YKD and at Anaktuvuk River both lost roughly 1.7 to 2.0 kg m$^{-2}$ of carbon (Mack et

al., 2011).  Carbon loss per area from these tundra wildfires is within the range of total above- and belowground
carbon loss from boreal wildfires, approximately 0.5 to 4 kg m$^{-2}$ (Walker et al., 2020a; Walker et al., 2018b; Rogers
et al., 2014), and close in value to the median wildfire loss of carbon, 2.54 kg m$^{-2}$, across all land cover types in
Alaska between 2001 and 2012 (Veraverbeke et al., 2015b).  Despite the additional combustion of overstory
vegetation in forests, carbon loss in both tundra and boreal systems comes primarily from belowground organic
matter (Walker et al., 2020a; Walker et al., 2018b; Rogers et al., 2014).  Compared to remote sensing-derived
estimates of combustion, our per unit area *in situ* measurements are on the low end but comparable to large-scale
means across Alaska between 2001 and 2018 (Fig. A9).  However, more representative field measurements in tundra
fires are needed to validate remotely sensed combustion measurements, as the currently available gridded products
(Potter et al. 2022; Scholten et al. 2021; Veraverbeke et al. 2017a; Veraverbeke et al., 2015a) are driven entirely by
measurements in boreal forests. Although data from the Anaktuvuk River and YKD fires shows carbon loss per unit
area can be similar between boreal and tundra wildfires, information on carbon emissions from tundra wildfires is
substantially more limited than boreal forest studies (He et al., 2021; Veraverbeke et al., 2021).

Given a fire return interval in the YKD of over around 200 years (Sae-Lim, 2019), fires there likely do not

burn through all the carbon accumulated since the last fire. However, tundra fires may initiate post-fire successional
trajectories, such as increased shrub cover, that may promote shorter fire return intervals (Hu et al., 2015; Rocha et
al., 2012). Shorter fire return intervals, in turn, gradually deplete the available soil carbon stocks. For example, He
et al. (2021) showed current tundra areas with frequent fire have relatively shallow soil organic layers. Moreover,
our radiocarbon data show that as fires burn deeper into the tundra, they release older carbon more rapidly because
the density of carbon by age increases. Coupling these burn-prone successional trajectories with projected increases
in tundra wildfire occurrence over the next century due to more frequent hot and dry conditions (Hu et al., 2015;
Rocha et al., 2012; Joly et al., 2012; Hu et al., 2010), as well as increases in lightning (Chen et al., 2021; Bieniek et
al., 2020; Veraverbeke et al., 2017b), tundra wildfires may begin to burn carbon of an older age than the historic fire
return interval. In this case, some tundra regions may transition into fire-driven carbon sources, similar to what has
been observed in North American boreal forests (Walker et al., 2019).

Immediate combustion is not the only impact wildfires have on tundra carbon stocks, nor do measures of

carbon loss fully describe the effect of wildfire emissions on the climate. Fires drive higher soil temperatures and
deeper active layer depths that can persist over multiple decades after fire (He et al., 2021; Hu et al., 2015; Rocha et
al., 2012), which is caused by the loss of insulating vegetation and upper soil layers from combustion (Rocha and
Shaver, 2011b). Higher soil temperatures and deeper active layer depths may catalyze permafrost degradation and
thermokarst development (Jones et al., 2015) that accelerates the effects of climate warming on tundra ecosystems.
Tundra fires also emit multiple gaseous and aerosol species, which have a net positive radiative forcing for at least
80 years post-fire. Hence, even if all of the carbon lost from fire in a tundra ecosystem were to be sequestered again
over time through increased photosynthesis, the long-term radiative impact of gaseous and aerosol emissions would
still generate atmospheric warming and thus positive climate feedbacks, which are amplified by the post-fire
permafrost thaw and degradation not accounted for in this study.

This positive radiative forcing is driven by sustained radiative forcings from long-lived GHGs, as well as

significant contributions from short-lived climate forcers (SCLFs) such as ozone, NMVOCs, and CO. Carbon
dioxide has the highest radiative forcing at both short and long time horizons because it is emitted in the highest
quantity (Akagi et al., 2011) and its concentration decreases relatively slowly over time, especially in the RCP 4.5
and 8.5 scenarios (Joos et al., 2013).  Aerosols and $NO_x$ are SCLFs with both positive and negative radiative
forcings due to distinct mechanisms for affecting the Earth's radiative balance.  Aerosols exert a direct radiative
forcing by reflecting or absorbing sunlight in the atmosphere and when deposited on high-albedo surfaces, mainly
snow and ice (Bond et al., 2011).  Although black carbon aerosols have a positive direct radiative forcing effect and
organic carbon aerosols have a negative direct radiative forcing effect, the magnitude of black carbon's radiative
forcing far exceeds that from organic carbon (Myhre et al., 2013) (note these are combined in our representation).
Indirectly, aerosols serve as the nuclei for cloud formation, which increases the Earth's albedo and exerts a negative
radiative forcing (Ward et al., 2012).  Nitrogen oxides have a near-term positive radiative forcing because they act in
concert with NMVOCs and CO to increase ozone concentrations, yet they have a long-term negative radiative
forcing because they decrease the lifetime of $CH_4$ (Collins et al., 2013).

We include a total radiative forcing with and without aerosols and ozone precursors, CO, NMVOCs, and

$NO_x$.  The radiative forcing of these ozone precursors and aerosols is uncertain within current literature, especially
when originating in the Arctic, because of their short atmospheric lifetimes and dependence on regional transport
patterns (Bond et al., 2011, Quinn et al., 2008).  However, the long-term cumulative radiative effect of these gaseous
emissions is relatively minimal in comparison to the long-lived GHGs and ozone, and both estimates of total
radiative forcing are of the same order of magnitude as previous estimates of the radiative forcing of gaseous
emissions from boreal forest fires (Huang et al., 2016; O'Halloran et al., 2012).

We found that as the predicted concentration of future emissions increases across RCP scenarios

(Meinshausen et al., 2011), the net radiative forcing of gaseous and aerosol emissions from the fires decreases
slightly.  Although counterintuitive, this result is due to the dependency of the radiative forcing equations for long-
lived GHGs, namely $CO_2$, $CH_4$, and $N_2O$, on the ambient concentration of a gas. In more extreme future climate
scenarios, the atmosphere generally has higher concentrations of these GHGs, meaning the additional quantity of gas
emitted by fire has a more diluted effect.  This effect is largely counter balanced by the fact that land and ocean $CO_2$
sink capacities are diminished in more extreme future climate scenarios, leading to longer atmospheric $CO_2$ lifetimes
(Fig. A10).  The net result is that tundra wildfires have a relatively similar positive radiative forcing under all future
emissions scenarios.
While the positive radiative forcing due to gaseous emissions we calculate for the 2015 YKD fire season is
within the range of previously calculated radiative forcings for gaseous emissions from boreal wildfires (e.g., Huang
et al., 2016, Randerson et al., 2006), boreal forests also have significant multi decadal post-fire albedo changes due
predominantly to the combustion of overstory trees that expose snow in the fall, winter, and spring (Lyons et al.,
2008). Randerson et al. (2006) and O'Halloran et al. (2012) show that the negative radiative forcing of these albedo
changes can exceed the positive radiative forcing of gaseous emissions and generate a net cooling effect for boreal
forest fires over long time horizons in specific locations. It should be noted, however, that (i) the net cooling or
warming is heavily dependent on combustion levels, pre-fire canopy composition, and time horizon; (ii) fire-induced
albedo forcings predominantly impact the regional climate (Rogers et al., 2013), as opposed to globally-mixed
GHGs; and (iii) this cooling impact is expected to diminish with future climate change due to decreases in spring
snow cover (Potter et al., 2020).
In contrast, tundra ecosystems lack an overstory, and hence fires generate relatively short-lived decreases in
albedo (Rocha et al., 2012, French et al., 2016). These albedo changes are only caused by char and the exposed soil
surface in the summer because tundra vegetation rarely exists above the snow layer (Mack et al., 2011). As a result,
within four years after the Anaktuvuk River fire, low post-fire surface albedo recovered to its pre-fire reflectance as
the vegetation grew over the charred soil surface (French et al., 2016; Rocha and Shaver, 2011b). The decreased
surface reflectance contributes to increases in soil temperature and thaw depth after fire (Rocha and Shaver 2011b).
Therefore, the positive radiative forcing caused by tundra wildfire gaseous and aerosol emissions is not offset by
post-fire albedo changes as seen in boreal forest fires. From 2001 to 2018 about 0.5 Mha of boreal forest burned on
average yearly compared to a yearly average of 0.017 Mha of tundra in Alaska (Scholten et al., 2021). Across the
entire Arctic region, about 9.0 Mha of boreal forest and 0.66 Mha of tundra burned on average yearly between the
years 1997 and 2016 (van der Werf et al., 2017; biome extents defined by Dinerstein et al., 2017). Given potentially
comparable per area emissions from boreal and tundra fires, boreal fires still release more carbon globally. Future
work could compare the total radiative forcing of tundra and boreal wildfires across their global domains, by
integrating gaseous, albedo, and other climate forcing effects for each biome. The predicted increase in frequency
of tundra wildfires, their lack of cooling from post-fire albedo, and their contribution to increased permafrost thaw
could drive a net radiative forcing higher than boreal wildfires, which in some cases may cool the climate (Oris et
al., 2014; O'Halloran et al., 2012; Randerson et al., 2006).
For the purpose of calculating the radiative forcing of wildfires in future research, we make our
computational code available on Github (see Code and Data Availability section). Our workflow is applicable to
wildfires in other arctic-boreal ecosystems, and it could be applied to any ecosystem globally with the adjustment of
emissions factors for the specific material burning (Akagi et al., 2011) and global warming potentials for SCLFs
based on the geographic region of the wildfire (Myhre et al., 2013). As a result, our method of calculating the
radiative forcing of wildfires can be used for studies across multiple regions with the appropriate parameters, which
can advance a more complete understanding of the effects of wildfires on the climate.
Future work could clarify some of the uncertainties that were not explicitly analyzed in our final estimates
of carbon loss and radiative forcing. We did not account for combustion of aboveground tall woody vegetation in
our estimates of carbon loss from the YKD fires. Although woody biomass is relatively sparse on the YKD
landscape, it represents an additional source of carbon from combustion (for boreal forest fires, see Walker et al.,
2020c). Despite the lack of difference we found between soil and vegetation matter characteristics across vegetative
reference point groups, there could have been variation in these characteristics outside of the primarily peat plateau
areas that we sampled. Burn depth estimated between *Dicranum* reference points was more variable (although not
significantly different) than burn depth estimated between *Sphagnum* reference points. This variation could indicate
more variable burn dynamics around *Dicranum* or better efficacy of *Dicranum* for measuring extreme burn depths,
which highlights the potential bias introduced by choosing specific combustion markers. Likewise, we eliminated
tussock measurements from our final analysis because they were relatively scarce on the landscape and yielded
inconsistent results, perhaps due to burn dynamics around tussocks that remain obscure using our combustion
measurement method. Minimizing potential bias by identifying combustion markers and soil characteristics for
different land cover types at a higher resolution within a fire, similar to how different combustion markers are used
regionally (Walker et al., 2020c; Mack et al., 2011; Boby et al., 2010) would further improve estimates of carbon
loss following wildfire in tundra systems.
Our assignment of low, moderate, and high severity was based on our particular collection of field
measurements and therefore may not translate well to other fire complexes. While dNBR is widely used to assess
fire severity, its accuracy in high latitude tundra systems may be diminished by environmental and methodological
factors, such as the timing of image selection and rapid post-fire greening (Chen et al., 2020a; Loboda et al., 2013).
Other remotely sensed products have shown promise in tundra (e.g., Chen et al., 2020b), but the applicability of
these results to other tundra systems remains to be explored.  Within low severity areas, there were likely patches of
unburned tundra, potentially leading us to overestimate organic matter and carbon loss measurements.  We also used
a relatively simple method for extrapolating emissions based on fire severity. More involved techniques such as
relating mechanism-based geospatial predictors to field plots using statistical techniques and machine learning,
which has been employed for boreal forests in Alaska and Canada (Dieleman et al., 2020; Walker et al., 2018b;
Veraverbeke et al., 2015b; Rogers et al., 2014), would likely yield more robust estimates for regional emissions.  In
this context, we strongly recommend increasing the number of tundra fire carbon emission observations in different
regions to facilitate synthesis and biome-wide modeling, as has been done for boreal forests (Walker et al., 2020c;
Walker et al., 2020b; Walker et al., 2020a).

Our radiative forcings model uses data and algorithms from multiple previous studies, including for

emissions factors (Akagi et al., 2011), RCP scenarios (Meinshausen et al,. 2011), GWPs (Myhre et al., 2013), ozone
and CO concentration ratios (Jaffe and Widger, 2012), and aerosol direct and indirect effects (Ward et al., 2012), all
of which may not be geographically, biologically, or temporally specific to the YKD fires. These factors are also
known to be variable depending on fuel types, fire severity and lifetime, and atmospheric dynamics at the time of
and following fire (e.g., Wiggins et al., 2016; Jaffe and Widger, 2012; Chen et al., 2009).  Since emissions factors
for tundra burning were not available, we used emissions factors for burning boreal vegetation.  Using boreal
emissions factors may overestimate emissions from woody vegetation in tundra and thereby increase the mass of
certain gasses released.  Similarly, boreal emissions factors might also misrepresent other tundra specific gaseous
emissions due to different soil and herbaceous vegetation compositions in the two biomes.  Future research could
integrate the range of variation associated with emission factors to quantify the uncertainty they introduce into
radiative forcing estimates and directly measure the emissions factors of biomass combustion in tundra ecosystems
for more accurate radiative forcing estimates.  We include estimates of radiative forcing with and without SCLFs,
because the radiative forcings of SCLFs emitted from the Arctic are uncertain in current literature.  Additional
studies assessing how regional atmospheric patterns, emission location, and abbreviated atmospheric lifetimes
govern the radiative forcing of SCLFs would constrain this uncertainty.  Furthermore, our model does not include
the potential mitigating or exacerbating effects of post-fire ecosystem changes on gas concentrations, such as
changes in post-fire vegetation (Frost et al., 2020; Zhou et al., 2019; Barrett et al., 2012), increased soil respiration
(Rocha and Shaver, 2011a), and increased active layer depths and permafrost degradation (Holloway et al., 2020;
Jafarov et al., 2013; Rocha et al., 2012).  With reparameterizations specific to individual fires, our model could be
made more accurate for future work.
**5 Conclusions**

To our knowledge, our study is the first to account for both carbon loss and radiative forcing from tundra

wildfires.  We develop a method that pairs *in situ* measurements of combustion levels from local pre-fire reference
points with remotely sensed burn severity data to scale organic matter and carbon loss to the entire fire area.  Our
estimate of the radiative forcing of gaseous and aerosol emissions from the fire event suggests that tundra wildfires
positively reinforce climate warming; however, the role of post-fire vegetative regeneration in mitigating these
carbon emissions remains unknown. Our results stress the importance of considering tundra wildfires in assessing
climate feedbacks and the need for future research that more explicitly discerns the warming effect of fires across
the tundra biome.

















**Appendix A**

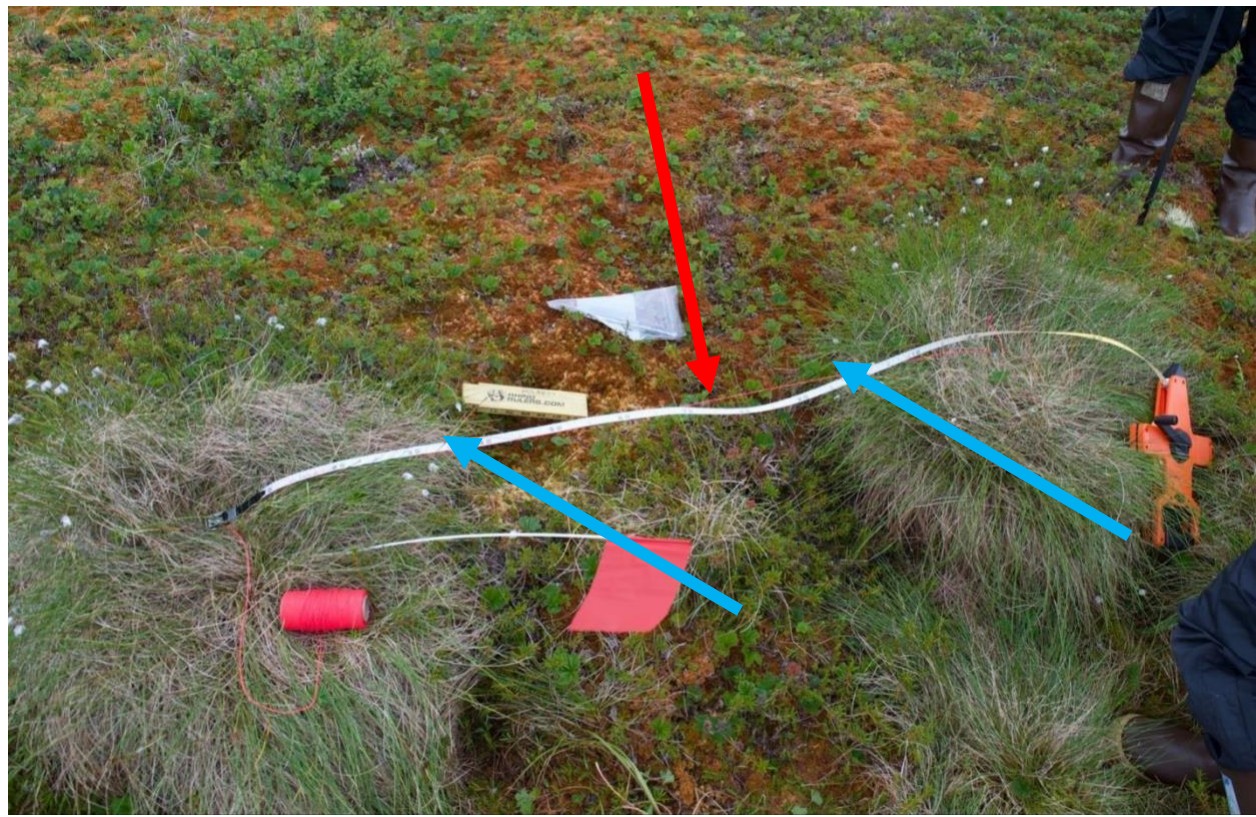

**Figure A1.** Height measurement of *Eriophorum vaginatum* in an unburned area from the taut string (red arrow) to
the dense vegetation layer between the starting and ending points (blue arrows) at the edge of each tussock patch.

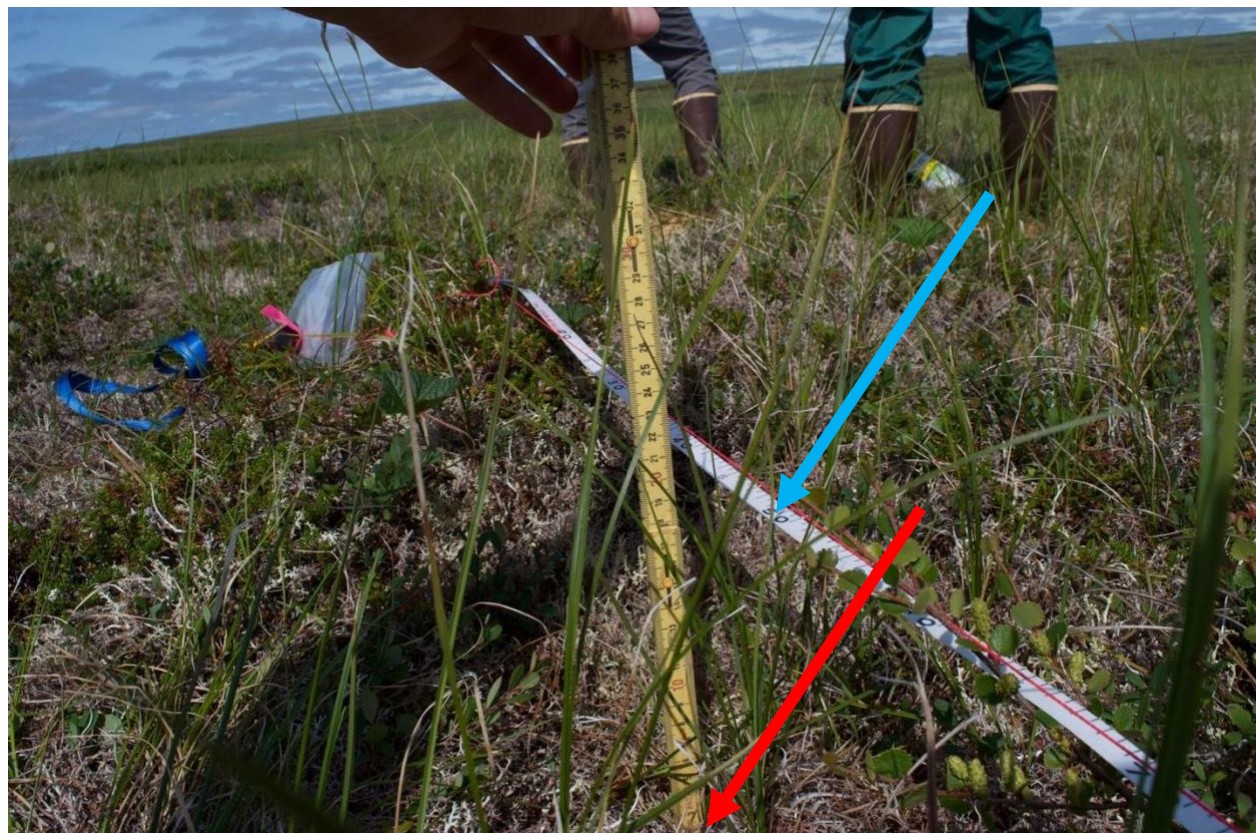

**Figure A2.** Measurement from the taut string to the dense vegetative layer (red arrow) in an unburned area every 25
centimeters (blue arrow).

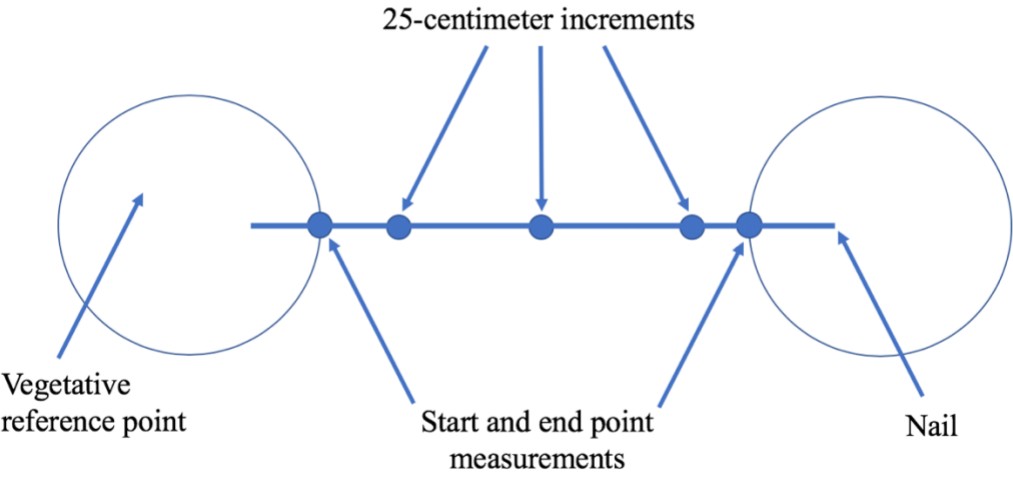

**Figure A3.** A schematic of the height measurements made along a transect between two vegetative reference points.

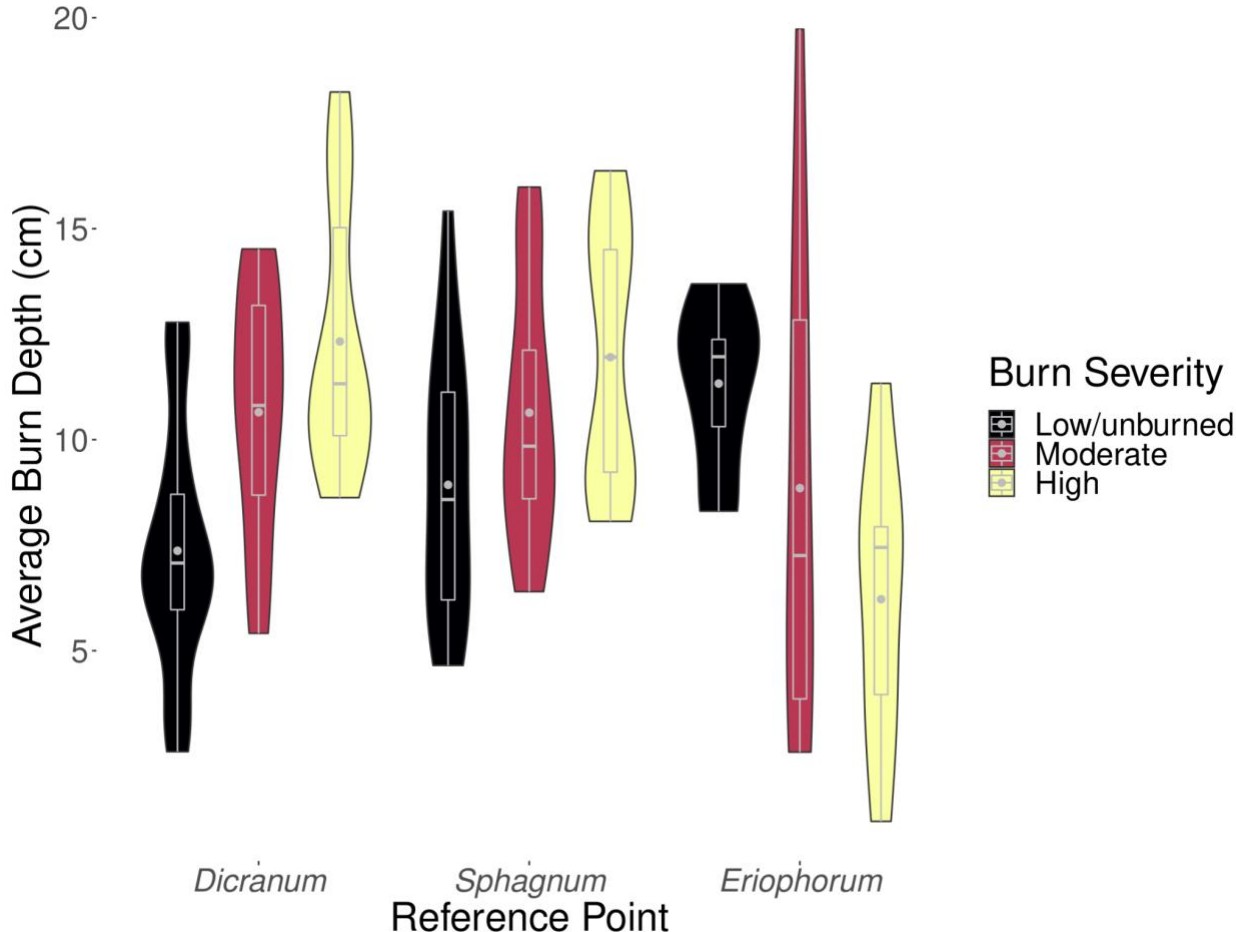

**Figure A4.** The distributions of average burn depth per transect within each reference point (including *Eriophorum*
*vaginatum*) across burn severity. Boxes encompass the middle 50% of data, whiskers are the upper and lower
quartiles, horizontal lines intersecting boxes show the median, and gray points are the mean.

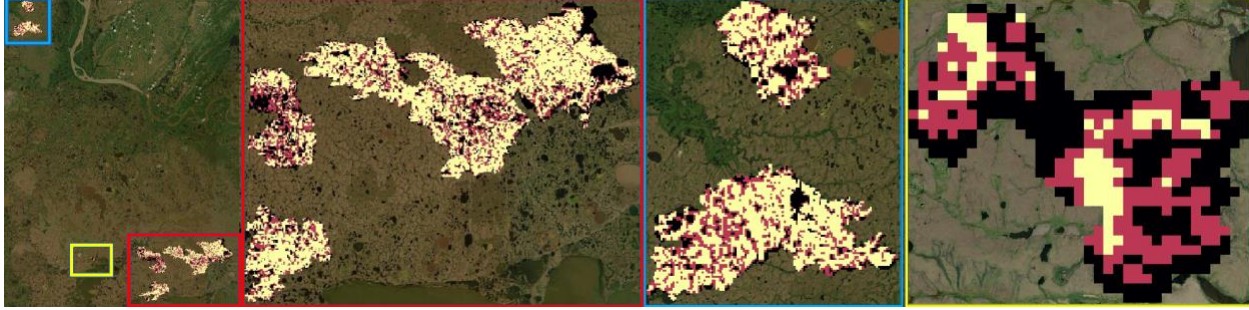


**Figure A5.** Fire severity across the YKD 2015 fire area categorized using dNBR values into low severity/unburned
(black), moderate severity (red), and high severity (yellow) categories. The panels show the total fire area (far left),
then close ups of the southeastern (red), northwestern (blue), and south-central (yellow) fire scars, respectively.
Basemap sources: Esri, DigitalGlobe, GeoEye, i-cubed, USDA FSA, USGS, AEX, Getmapping, Aerogrid, IGN,
IGP, swisstopo, and the GIS User Community.

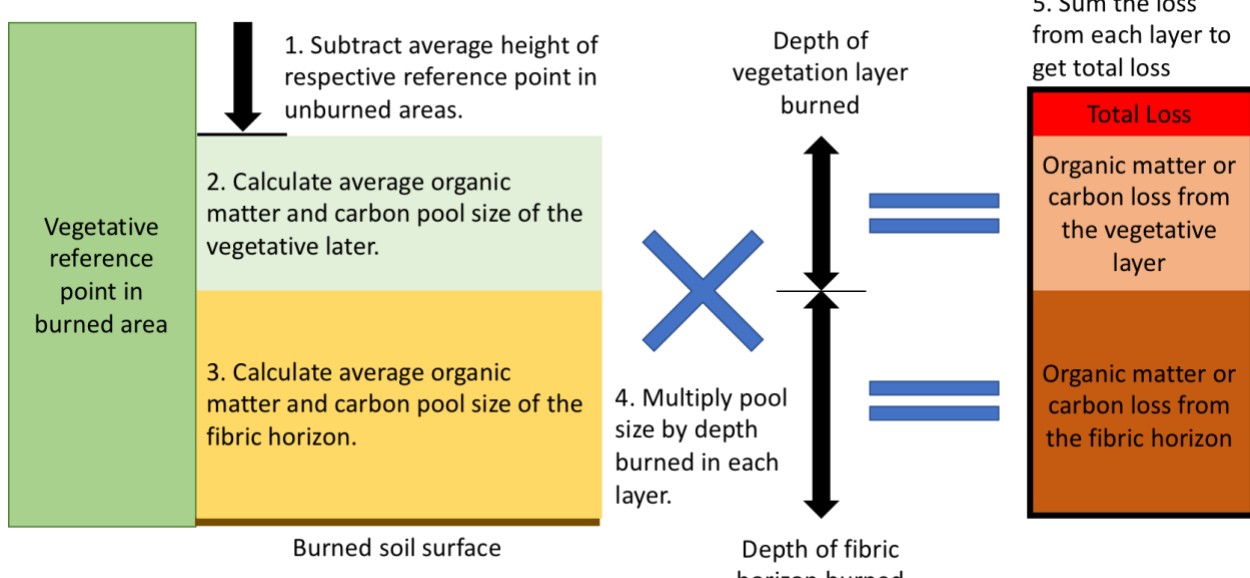

**Figure A6.** A schematic of organic matter and carbon loss calculations at a single transect in the burned area.

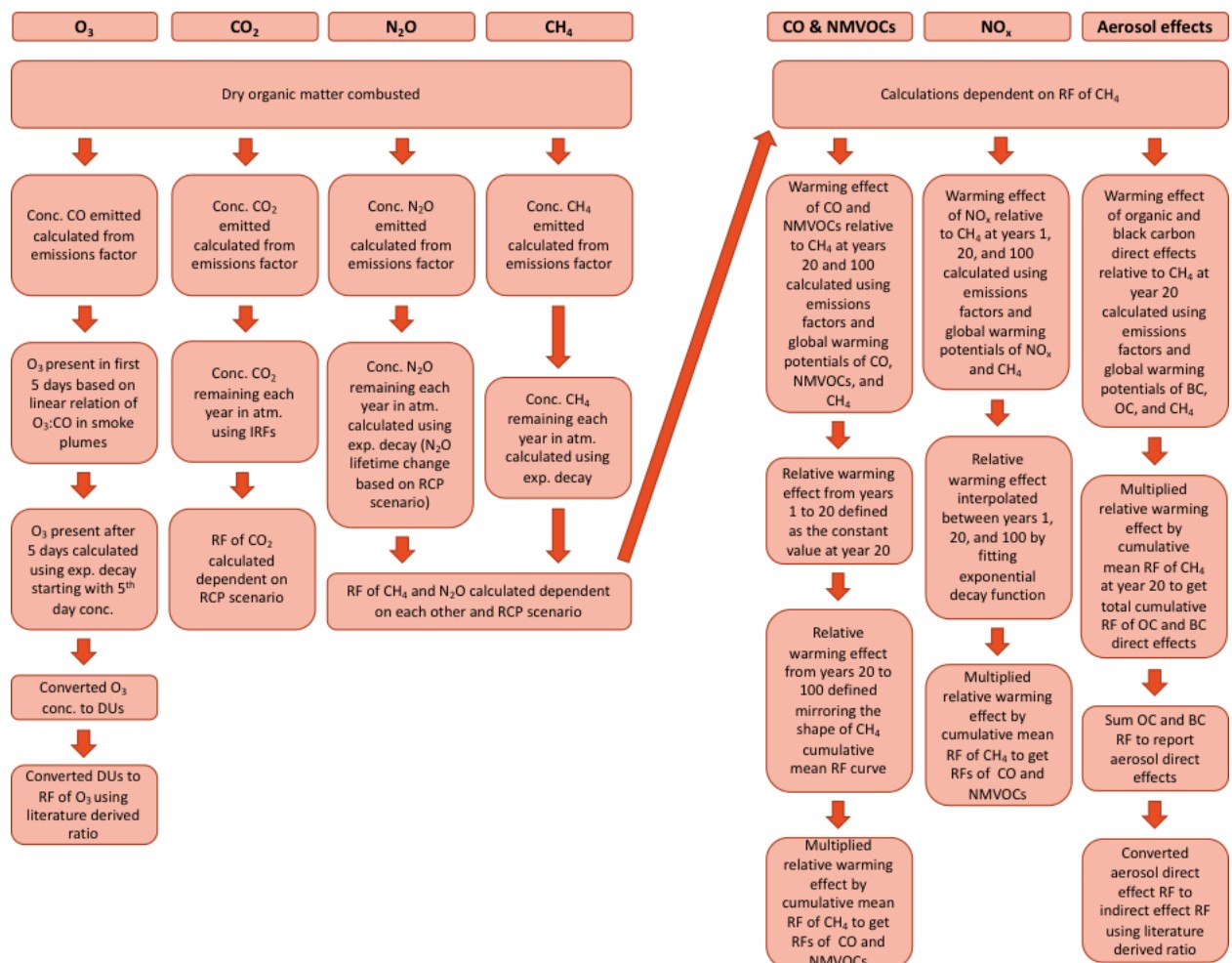

**Figure A7.** Flowchart of the methodology for the calculation of the radiative forcings of gaseous emissions from
biomass combustion generally applicable to any biome with adjustment of biome-specific parameters.

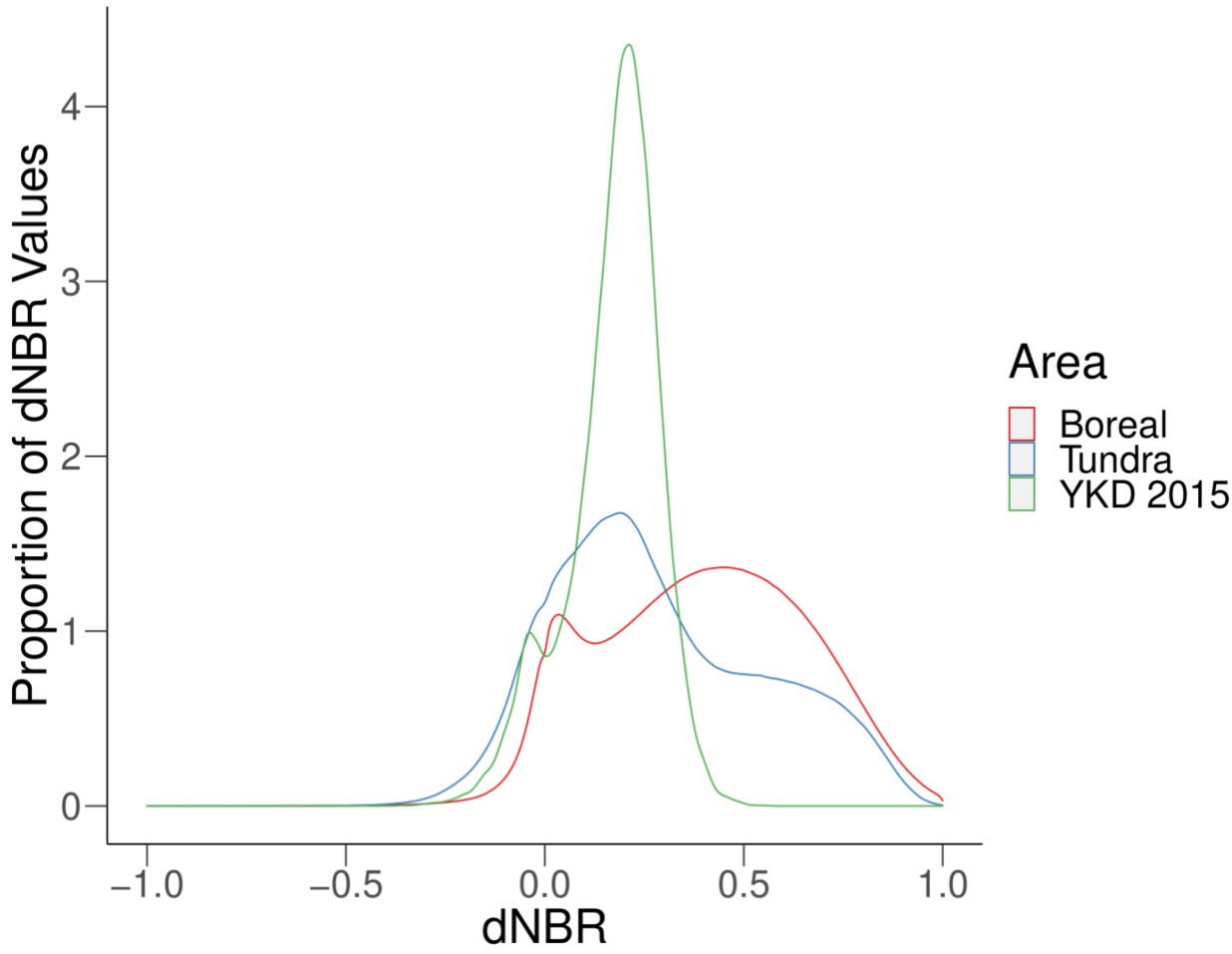


**Figure A8.** The distribution of dNBR values for the 2015 YKD fire season compared to all fires in boreal and tundra systems in Alaska between the years 1989 and 2019.

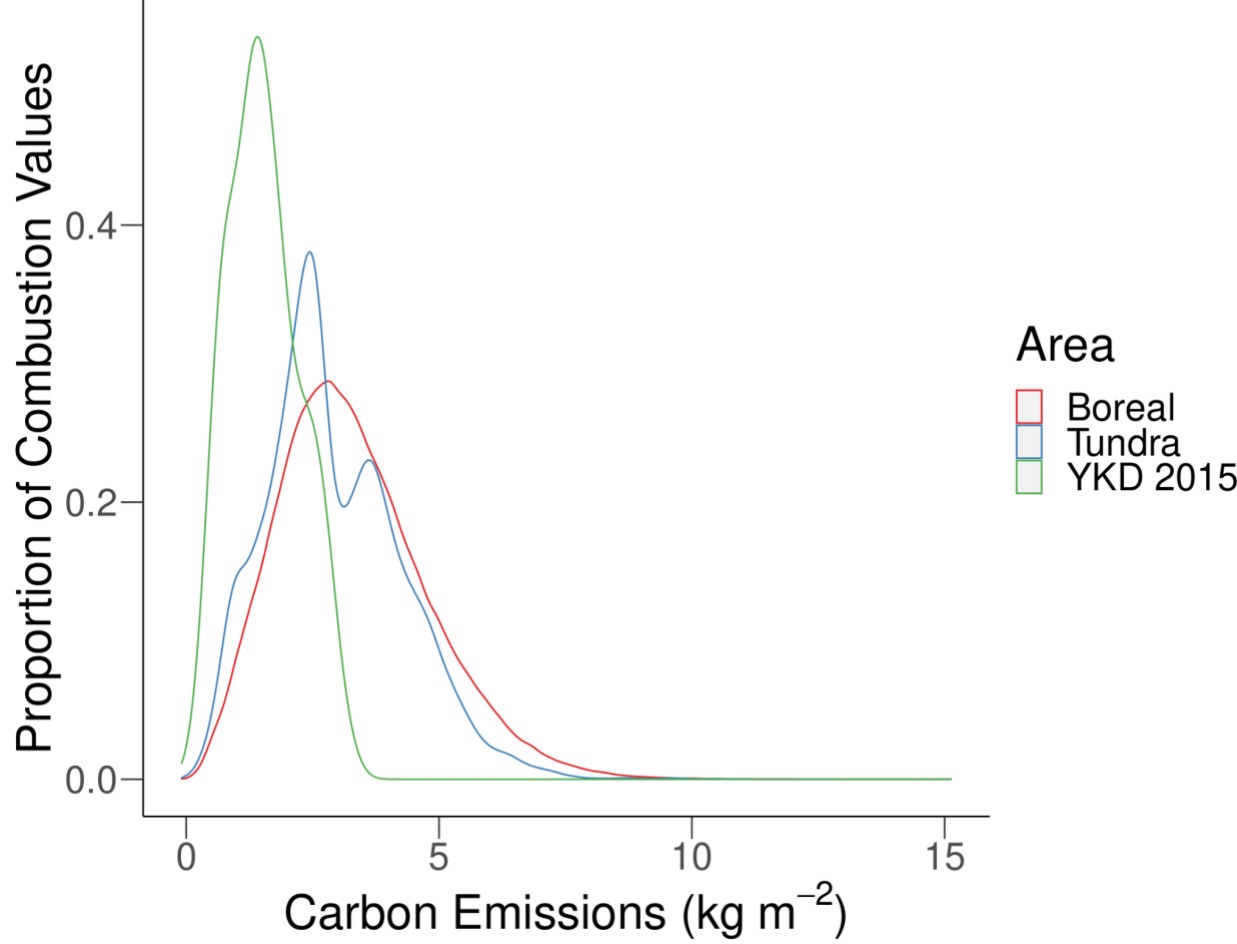

**Figure A9.** The distribution of carbon emission values from transect measurements in the YKD 2015 fires reported
herein and from remotely sensed pixels of all fires in boreal and tundra systems in Alaska between the years 2001
and 2018 (Scholten et al., 2021).

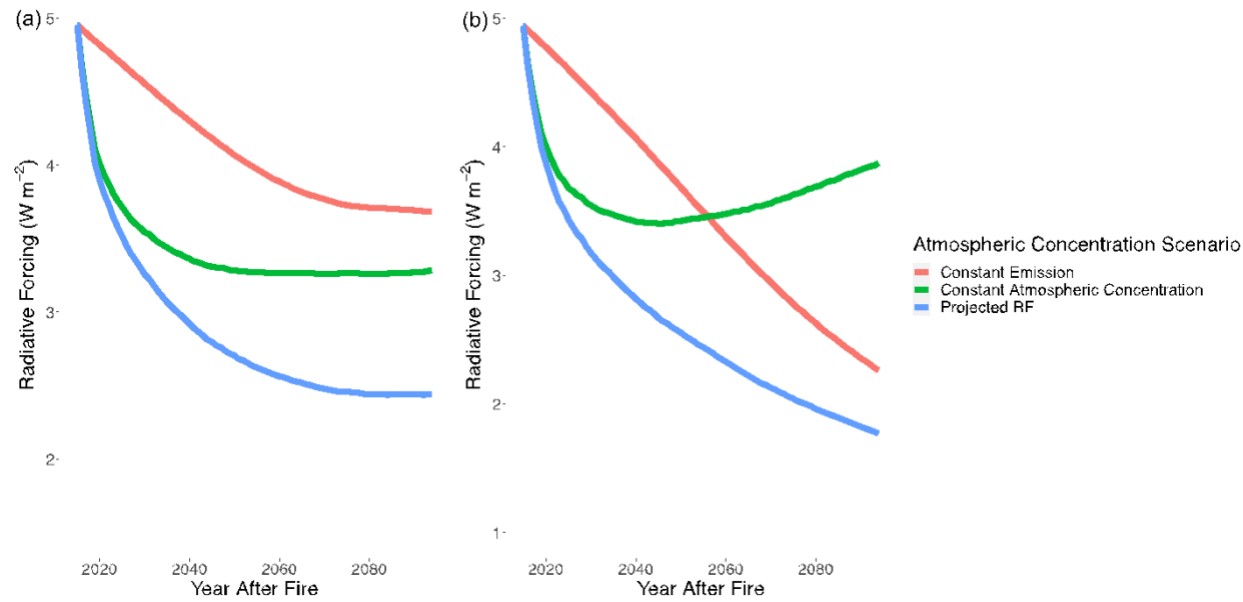

**Figure A10.** The radiative forcing of $CO_2$ emitted from the 2015 YKD fire season for RCP 4.5 (a) and 8.5 (b) when (i) holding the emission constant (red lines) so that no emitted $CO_2$ is sequestered, (ii) holding the background atmospheric concentration constant (green lines), and (iii) the combination of these, resulting in our main $CO_2$ radiative forcing equation (blue lines; Eq. 7). In the case of (i, red lines), increasing atmospheric $CO_2$ concentrations dampen the radiative impact of fire-emitted $CO_2$. In the case of (ii, green lines), the only time-varying impact on $CO_2$ radiative forcing is ocean and land sinks, which become less effective and even reverse with more extreme climate change according to model results presented in Joos et al. (2013). These lines are similar to the impulse response function of $CO_2$ for their respective scenarios in Joos et al. (2013).

**Code and Data Availability**

The radiative forcings calculation code is available on GitHub and Zenodo via DOI: 10.5281/zenodo.6617455

(Moubarak, 2022). All field and lab raw data used in this study, including vegetative reference point height

measurements and soil and vegetation core metrics, and combustion data products derived from our calculations are

publicly available in the NSF Arctic Data Center via DOI:10.18739/A2PR7MV5P (Moubarak et al., 2020).

**Author Contributions**

MM, BR, SS, and SN were involved with conceptualization, funding acquisition, and methodology. SP completed

investigation of burn severity data and jointly completed visualization with MM. MM, SS, and BR did investigation

of burn depth measurements and soil and vegetation characteristics. MM analyzed data and developed the radiative

forcings model. SS and SN did project administration for the field sampling. MM curated data and wrote the

original manuscript draft. MM, BR, SS, SP, and SN reviewed and edited the final published work.

**Competing Interests**

The authors declare that they have no conflict of interest.

**Acknowledgements**


This work was funded by the National Science Foundation (NSF-1915307 to SMN, NSF-1915307 to SS),

Casstevens Family Fund (to MM), and the National Aeronautics and Space Administration (NASA) Arctic-Boreal
Vulnerability Experiment (ABoVE; NNX15AU56A to BR) and the Gordon and Betty Moore Foundation (8414 to
SMN and BR).  The authors would like to thank Dr. William Pfitsch for his helpful mentorship in the synthesis of
this project and creation of the written work.  Also, the authors would like to thank Dr. Catherine Beck for offering
her knowledge about radiocarbon sample preparation and analysis.

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
