# Peer review of "Carbon emissions and radiative forcings from tundra wildfires in the Yukon-Kuskokwim River Delta, Alaska"

_Biogeosciences, 2022_

## Author Comment (AC1)

**Referee #1**

Wildfires play an important role in the carbon cycle and Earth's energy balance. Despite increasing incidences of tundra wildfires, very few studies on tundra-wildfire carbon emissions exist, and the radiative forcing associated with these emissions is not available. The manuscript of Moubarak et al. attempts to estimate carbon emissions and the resultant radiative forcings from tundra wildfires in the Yukon-Kuskokwim River Delta (YKD), Alaska. While it shows potential scientific significance to the climate community, I find it hard to read overall. It is unclear to me how the authors estimate the carbon emissions and radiative forcing, and the current estimate of radiative forcing is not convincing to me either.

RESPONSE: Thank you for your helpful feedback and the chance to clarify our methodology and importance of our results. Below we have responded to your general and line-specific comments individually.

General comments

1. The manuscript fails to show the total global amounts of carbon emissions from tundra wildfires and how significant they are, for example, compared to those from the boreal forest on a global basis. While the current estimate of emissions from the YKD area within Alaska is essential, it is not clear whether the global emissions are still significant from the global perspective, given that the Tundra ecosystem covers only a small fraction of the earth.

RESPONSE: Given that tundra fires burn less area than boreal fires, with potentially comparable per unit area emissions, they contribute less to global carbon emissions than boreal fires. We will include a statement making this comparison in the discussion, where we cite the relative burned area for tundra and boreal fires in Alaska between the years 2001 and 2018. During this time frame, there was a yearly average of 0.5 Mha of boreal burned area compared to 0.017 Mha of tundra (Scholten et al., 2021). Despite the smaller burned area, tundra fires play an important role in ecosystem function and global climate because they can contribute to permafrost thaw, lack post-fire cooling effects from albedo changes seen in boreal fires, and are increasing in frequency. These factors heighten their importance in climate change. We will highlight these effects with our statement about global emission totals.

Scholten, R.C., S. Veraverbeke, R. Jandt, E.A. Miller, and B.M. Rogers.: ABoVE: Ignitions, Burned Area, and Emissions of Fires in AK, YT, and NWT, 2001-2018, ORNL DAAC, Oak Ridge, Tennessee, USA [data set], https://doi.org/10.3334/ORNLDAAC/1812, 2021.

2. The radiative forcing estimation in the current format looks faulty. First, the radiative forcing is given as a global average. However, the emissions and concentrations used here are the regional mean values. In addition, how do the authors translate the emissions to the concentrations of the gas/aerosol remaining in the atmosphere?

RESPONSE: Calculating global averages is standard practice for radiative forcing modeling of regional wildfires (e.g., Huang et al., 2016, O'Halloran et al., 2012, and Randerson et al., 2006). We will more explicitly state our method's precedent with these citations in the methods section, '2.7: Radiative forcings model'. Our final estimate for radiative forcing from this tundra wildfire season was bounded by estimates of the gaseous radiative forcing from boreal wildfires (Huang et al., 2016, Randerson et al., 2006). Thus our calculation is within reason, especially

given the similar levels of per unit area combustion we show between the fire season we measure and boreal fires. We will more explicitly contextualize our results with this comparison in the discussion by stating that the radiative forcing we calculated for these tundra wildfires is within the range of estimates for the gaseous radiative forcing of boreal fires.

We translated from emissions to atmospheric concentration of gaseous species remaining by converting from a mass emitted (calculated using emissions factors) to a volume as a molar fraction of the atmosphere. This step was crucial in converting from a regional emission to a global value, and we will make it more clear in our methods in response to the reviewer's line-specific suggestions below. Such a conversion was possible and is standard because greenhouse gasses are generally long-lived and globally-mixed, and global averages allow for common comparison.

Huang, S., Liu, H., Dahal, D., Jin, S., Li, S. and Liu, S.: Spatial variations in immediate greenhouse gases and aerosol emissions and resulting radiative forcing from wildfires in interior Alaska, Theoretical and applied climatology, 123, 581-592, https://doi.org/10.1007/s00704-015-1379-0, 2016.

O'Halloran, T. L., Law, B. E., Goulden, M. L., Wang, Z., Barr, J. G., Schaaf, C., Brown, M., Fuentes, J. D., Göckede, M. and Black, A.: Radiative forcing of natural forest disturbances, Global Change Biol., 18, 555-565, https://doi.org/10.1111/j.1365-2486.2011.02577.x, 2012.

Randerson, J. T., Liu, H., Flanner, M. G., Chambers, S. D., Jin, Y., Hess, P. G., Pfister, G., Mack, M. C., Treseder, K. K. and Welp, L. R.: The impact of boreal forest fire on climate warming, Science, 314, 1130-1132, https://doi.org/10.1126/science.1132075, 2006.

Specific comments

Line 57: Is there anything to do with "more variable precipitation" here? How is it related to the "increasing temperature"?

RESPONSE: By mentioning more variable precipitation in the context of increasing temperature, we meant to highlight the more frequent co-occurrence of hot and dry conditions, which provides context for the climate and environmental drivers of increasing wildfires in the following paragraph. We will add the phrase "hence more frequently pairing hot and dry conditions" to make this explicit.

Line 164: How do the authors "maximize" physical separation of the unburned sites?

RESPONSE: We will remove the word "maximize" since it implies a formal optimization. Instead, we will cite the minimum distance of 1 kilometer between unburned sites.

Line 190-191: How do the authors calculate the organic matter combustion and carbon loss? I cannot get any hint from Fig. A4. Can they provide an equation and/or diagram to help readers understand this?

RESPONSE: Referencing Fig. A4 here was a typo, since it was meant to illustrate the trend in tussock burn depth measurements. We will move our reference of the figure to a later sentence

in the paragraph where we discuss our exclusion of tussock measurements because their burn depths correlated negatively with fire severity.

We also will add a schematic to the appendix to illustrate our calculation of organic matter and carbon loss at each burn depth transect. This figure will be referenced after text that describes how organic matter and carbon loss were calculated. Below is a sample of what that appendix figure will look like.

[Figure]

Line 192: Can the authors explain the reasons for the unexpected behaviors of tussock measurements?

REPONSE: Given the limited sample size of tussocks on the landscape and the lack of an explicit experimental design for this specific purpose, we cannot make definitive statements regarding the behavior of tussock measurements. We will suggest that tussocks may alter burn dynamics locally, while emphasizing that no conclusions can be drawn from our current dataset.

Line 246-248: How about the uncertainty associated with the emission factor?

RESPONSE: Since Akagi et al. (2011) combines measurements of emissions factors for each gaseous species from multiple studies with different designs, such as laboratory, field, and airborne, uncertainty is introduced into the emissions factor by the measurement technique. Post-emission chemical reactions from the released gaseous species cause the concentrations of specific gasses to vary after emission. As a result, how the gasses are measured might affect the emission factor. We will summarize this uncertainty in the methods.

Akagi et al. (2011) provides an estimate of natural variation for most emissions factors, which we do not include in our model. We also will highlight this in the uncertainty section of our discussion by stating future models might incorporate the natural variation of each emission factor to bound their estimates.

Akagi, S. K., Yokelson, R. J., Wiedinmyer, C., Alvarado, M. J., Reid, J. S., Karl, T., Crounse, J. D. and Wennberg, P. O.: Emission factors for open and domestic biomass burning for use in atmospheric models, Atmospheric Chemistry and Physics, 11, 4039-4072, https://doi.org/10.5194/acp-11-4039-2011, 2011.

Line 277-283. How do the authors get the atmospheric concentrations of CH4 and N2O from the fire emissions?

RESPONSE: While we mentioned briefly on line 250 (original submission) in our general description of the radiative forcings model that we converted the mass of the gaseous species emitted to volumes as molar fractions of the atmosphere, we will rearrange our text to make this methodology clearer for the gaseous species to which it was applied.  First, we will remove that original sentence on line 250 (original submission).  Then, for $CH_4$ and $N_2O$, we will add a sentence in section 2.8 stating that the concentration of both gasses was calculated by converting the initial mass emitted to a volume as a molar fraction of the atmosphere.  We also originally labeled $C_o$ in equation 3 as an "initial pulse mass of the gas."  Here we will change "mass" to "concentration."  For $CO_2$, we will add a similar sentence describing the conversion of mass to atmospheric concentration in section 2.9.   Finally, the same method was also used for CO when calculating the concentration of tropospheric ozone.  Therefore, we will add the same description for CO in section 2.10.

Line 294. Again, how do the authors estimate the CO2 concentration from the fire emissions?

RESPONSE: See response to line 277-283.

Line 318. Although ozone precursors can affect CH4 formation, how are aerosols related to CH4?

RESPONSE: Our original text on this line was misleading.  While aerosols are not explicitly related to CH4 formation, we chose to model them using methods similar to those we used for ozone precursors for the sake of consistency in GWP-derived radiative forcings calculations.  We will change the text to reflect this methodological choice by stating:

> "We derived the radiative forcings for ozone precursors based on $CH_4$ because the radiative forcing of ozone precursors is through their effect on methane over the long-term (Collins et al., 2013). To remain consistent with our methodology for ozone precursors, we derive the radiative forcings for aerosols from $CH_4$ as well."

Collins, W. J., Fry, M. M., Yu, H., Fuglestvedt, J. S., Shindell, D. T. and West, J. J.: Global and regional temperature-change potentials for near-term climate forcers, Atmospheric Chemistry and Physics, 13, 2471-2485, https://doi.org/10.5194/acp-13-2471-2013, 2013.

Line 340. Why is R20 used here? So only accounts for the cumulative radiative forcing at the 20-year time zone?

RESPONSE: Our original text did not fully explain our assumptions for our aerosol calculations.  We will rearrange the paragraph and add text to clarify.  R20 is used because the total radiative forcing of aerosols likely happens within the first year of the fire event, as aerosols

exit the atmosphere on a time scale of weeks.  Therefore, the cumulative radiative forcing of aerosols every year after the fire event is constant.  We chose to use the GWP of the aerosols at year 20 to calculate the radiative forcing and assumed this value was the same across all years.  We will change the text to state that:

> "The radiative effect of aerosol emissions happens within a year of the fire event, as fire aerosols are typically removed from the atmosphere via wet and dry deposition within a matter of weeks (Bond et al., 2011, Quinn et al., 2008).  We assumed that the cumulative radiative forcing of aerosols at any year after the fire event would be constant and equal to the radiative forcing of aerosols in the year of the fire event."

Bond, T. C., Zarzycki, C., Flanner, M. G. and Koch, D. M.: Quantifying immediate radiative forcing by black carbon and organic matter with the Specific Forcing Pulse, Atmospheric Chemistry and Physics, 11, 1505-1525, https://doi.org/10.5194/acp-11-1505-2011, 2011.

Quinn, P. K., Bates, T. S., Baum, E., Doubleday, N., Fiore, A. M., Flanner, M., Fridlind, A., Garrett, T. J., Koch, D. and Menon, S.: Short-lived pollutants in the Arctic: their climate impact and possible mitigation strategies, Atmospheric Chemistry and Physics, 8, 1723-1735, https://doi.org/10.5194/acp-8-1723-2008, 2008.

Line 350: what is the "gls" function?

RESPONSE: To make clear the "gls" function is defined by the nlme package, we will rearrange the sentence to say, "we used the nlme function "gls" to fit a linear model."  We will also clarify that the model was fit using the generalized least squares method.

Line 359: what is the "lme" function?

RESPONSE: Similarly, to emphasize the "lme" function came from the nlme package, we will add the phrase "defined in the nlme package."  We will also clarify that the model is fit using the restricted maximum likelihood method.

Line 396: what is "SEM"?

RESPONSE: SEM stands for standard error of the mean, and the text will be changed to reflect this definition.

---

## Author Comment (AC2)

**Referee #2**

The following comments pertain to the manuscript "Yukon-Kuskokwim River Delta 2015 fire burn depth measurements and unburned soil and vegetation organic matter and carbon content collected in 2019". Overall, I think the findings presented in this manuscript are of high scientific importance and I applaud the authors for conducting this work. Having said that, I have a few major issues with this manuscript. The first one concerns the language of this manuscript. If my understanding of this manuscript is correct, the findings presented here stemmed from field data collected in a single burn scar in YKD. While these findings are important in their own rights, the authors (mostly in the results and discussion sessions) made a lot of statements that are obviously overly strong. Tundra wildfires, while being overall understudied, vary substantially based on climate, terrain, species composition, etc. As a result, while this study shows that the YKD fire that it looked into had a similar emission rate to the Anaktuvuk River Fire (ARF), they still represent two cases out of many more tundra fires, many of which are much less severe than the ARF. Therefore, the authors should be very careful when they make statements concerning "tundra fires". You can indicate that the YKD fire and the ARF likely represent the upper level of the impacts of tundra fires (in terms of carbon emission and radiative forcing), but you can't just hint that these are representative situations.

My second major issue with this manuscript is the fact that while it has a substantial discussion session, it fails to discuss the sources of uncertainty introduced by the methods that were used. For example, this study used dNBR as the indicator for burn severity, however, there have been studies that show that dNBR may not be a very good burn severity indicator in the tundra (Loboda et al 2013) and that there may be better indicators (Chen et al 2020). Another major potential source of uncertainty stems from the fact that the emission factors the authors used were from boreal fires, which as the authors state in the method section may lead to an overestimation. However, this overestimation wasn't further discussed in the discussion.

Additionally, in terms of dNBR, there have been a lot of studies showing the caveats associated with dNBR when it is used in high northern latitude settings (eg, Chen et al 2020). The author failed to take these into account completely. Additionally, the authors never showed the audience how the YKD fire that they picked compared with other tundra fires. This is a piece of critical information that is needed for a wider audience to understand the scientific importance of your work.

Loboda, T. V., French, N. H., Hight-Harf, C., Jenkins, L., & Miller, M. E. (2013). Mapping fire extent and burn severity in Alaskan tussock tundra: An analysis of the spectral response of tundra vegetation to wildland fire. Remote Sensing of Environment, 134, 194-209.

Chen, Yaping, Mark Jason Lara, and Feng Sheng Hu. "A robust visible near-infrared index for fire severity mapping in Arctic tundra ecosystems." ISPRS Journal of Photogrammetry and Remote Sensing 159 (2020): 101-113.

Chen, D., Loboda, T. V., & Hall, J. V. (2020). A systematic evaluation of influence of image selection process on remote sensing-based burn severity indices in North American boreal forest and tundra ecosystems. ISPRS Journal of Photogrammetry and Remote Sensing, 159, 63-77.

RESPONSE: Thank you for your helpful feedback, as it allows us to more accurately represent the scope of our findings and their importance to the scientific community. While our initial text generalized our results to all tundra fires, we will both scale back the language of our results and provide context for the fire season we measured compared to other fires in Alaska. First, we will clarify that our results represent combustion from a single tundra fire season in Alaska and may not be representative for all tundra wildfires. Instead of generalizing our results to all tundra fires, we will state that on a per unit area basis, the fire season measured herein, the ARF fire, and boreal fires have relatively similar levels of combustion, which suggests emissions rates per unit burned area from tundra fires can be comparable to boreal fires. These changes will be made in the discussion and in response to your line-specific comments below.

In addition, we will compare per unit area combustion from our study with all fires between 2001 and 2018 in boreal and tundra systems in Alaska, estimated via remote sensing and geospatial machine learning techniques, by adding a density plot to the appendix. A sample of that plot is given below. While field measurements of tundra combustion are currently lacking, but necessary to validate remotely sensed combustion products, our field measurements fell within the range of remotely sensed combustion in tundra and boreal systems. We will make note of this trend in the text by stating that:

> "Compared to remotely sensing-derived estimates of combustion, our per unit area *in situ* measurements are on the low end but comparable to large-scale means across Alaska between 2001 and 2018 ([Figure Below]). However, more representative field measurements in tundra fires are needed to validate remotely sensed combustion measurements, as the currently available gridded products (Potter et al. 2022; Scholten et al. 2021; Veraverbeke et al. 2017a; Veraverbeke et al., 2015a) are driven entirely by measurements in boreal forests."

In addition to adding this context in the discussion, we will add context for our fire season's size and dNBR values in the methods, results, and discussion sections. The 2015 YKD fire season burned roughly three times the annual average tundra burned area in Alaska between the years 2001 and 2018 but accounted for only a little more than half of the Alaskan tundra area burned in the year 2015 alone. Within the YKD, the 2015 fire season burned more area than any other year between 2001 and 2018. Most years in that time frame burned less than 10,000 hectares. In comparison, the 2015 YKD fire season burned 54,154 hectares. This context will be added in the first paragraph of the discussion. The dNBR values for the fire season we studied were intermediate compared to dNBR values for tundra and on the low side but within the range of values for boreal fires in Alaska generally. These contextual changes will clarify that while definitive conclusions about how our focal fires translate to other tundra fires cannot be made presently, the results of our study highlight the need for more attention to tundra wildfires from a climate perspective. We elaborate more on these contextual changes in response to your line-specific comment below as well.

We will expand the discussion section describing sources of uncertainty introduced by our methodology to add more description of the sources you mention here. First, we will describe that despite dNBR being the most widely used metric to assess fire severity, its accuracy may be diminished in high latitude tundra systems by environmental and methodological

factors. We appreciate the studies you suggested, and we will cite them in our manuscript. In the text, we will note that other metrics have shown promise in tundra systems, but how these results translate to other tundra systems remains to be explored. Second, throughout the manuscript we will note that our low severity fire class likely includes unburned patches of tundra, leading us to potentially overestimate combustion in low severity areas. Finally, we will reiterate in the discussion that since emissions factors for tundra fires are not available, we used boreal emissions factors, potentially leading us to overestimate gaseous emissions from woody vegetation. As noted in the discussion, our model was parameterized with data from various sources and is publicly available via GitHub. Since data on tundra wildfires is lacking to date, our parameterization could only approximate tundra fires. However, as these parameters are updated for tundra systems in future literature, our model can be updated. In addition to this general response, we address your detailed comments individually below.

[Figure]

References for general comment:

Potter, S., Veraverbeke, S., Walker, X. J., Mack, M. C., Goetz, S. J., Baltzer, J. L., Dieleman, C., French, N. H. F., Kane, E. S., Turetsky, M. R., Wiggins, E. B., and Rogers, B. M.: ABoVE: Burned Area, Depth, and Combustion for Alaska and Canada, 2001-2019, ORNL DAAC, Oak Ridge, Tennessee, USA [data set], https://doi.org/10.3334/ORNLDAAC/2063, 2022.

Scholten, R.C., S. Veraverbeke, R. Jandt, E.A. Miller, and B.M. Rogers.: ABoVE: Ignitions, Burned Area, and Emissions of Fires in AK, YT, and NWT, 2001-2018, ORNL DAAC, Oak Ridge, Tennessee, USA [data set], https://doi.org/10.3334/ORNLDAAC/1812, 2021.

Veraverbeke, S., Rogers, B. M., and Randerson, J. T.: CARVE: Alaskan Fire Emissions Database (AKFED), 2001-2013, ORNL DAAC, Oak Ridge, Tennessee, USA [data set], https://doi.org/10.3334/ORNLDAAC/1282, 2015a.

Veraverbeke, S., Rogers, B. M., Goulden, M. L., Jandt, R., Miller, C. E., Wiggins, E. B., and Randerson, J. T.: ABoVE: Ignitions, burned area and emissions of fires in AK, YT, and NWT, 2001-2015, ORNL DAAC, Oak Ridge, Tennessee, USA [data set], https://doi.org/10.3334/ORNLDAAC/1341, 2017a.

Here are my more detailed comments.

Line 38 "AK": Spell out Alaska.

RESPONSE: Will change.

Line 64 "North slope": North Slope

RESPONSE: Will change.

Line 97: More information about how this particular fire compares with other tundra fires (both from YKD and other tundra regions in Alaska) is needed (in terms of size, overall dNBR, etc).

RESPONSE: We will add a few pieces of information to put the YKD 2015 tundra fire season in context. We will add this information throughout the methods, results, and discussion where it is appropriate. First, as stated in response to your general comment, we will make comparisons based on burned area in the first paragraph of the discussion. The 2015 YKD fire season burned roughly three times the average annual burned area in Alaskan tundra between the years 2001 and 2018 but accounted for only a little more than half of the tundra area burned in Alaska in the year 2015. The 2015 fire season had the largest burned area within the YKD compared to the years 2001 to 2018. Most years in that time frame had burned areas less than 10,000 hectares. In comparison, the 2015 YKD fire season burned 54,154 hectares.

Also, we will compare the dNBR values from our study to dNBR values for all fires in Alaskan tundra and boreal systems. Differenced NBR for the fire season in this study was intermediate to dNBR for all tundra fires in Alaska from 1989 to 2019 and on the low side but within the range of dNBR for Alaskan boreal fires during this same timeframe. We will add our means of obtaining this information to the methods and the data will be cited in the results and as a density plot in the appendix. A sample of that plot is shown below. As mentioned in our general comment, we will compare the per unit area combustion from our focal fires to other remotely sensed per unit area combustion measurements for tundra fires in Alaska.

[Figure]

Line 105: Where did you get the burn scars? This needs to be specified.

RESPONSE: The burn scars were found using the Alaska Large Fire Database (Kasischke et al., 2002).  We will mention this in the current section 2.1 and the methodology section 2.4 that describes fire severity.

Kasischke, E. S., Williams, D. and Barry, D.: Analysis of the patterns of large fires in the boreal forest region of Alaska. International Journal of Wildland Fire, 11, 131–144, https://doi.org/10.1071/wf02023, 2002.

Line 181 dNBR's full name has already been given previously.

RESPONSE: We will replace the full name with the acronym.

Line 181 "Key and Benson 2006": This reference should be provided when dNBR was first mentioned.

RESPONSE: We will add this citation where dNBR is first mentioned in the introduction.

Line 183: CFMASK algorithm: this needs a citation.

RESPONSE: We will add the citation Foga et al. (2017).

Foga, S., Scaramuzza, P. L., Guo, S., Zhu, Z., Dilley, R. D., Beckmann, T., Schmidt, G. L., Dwyer, J. L., Joseph Hughes, M., and Laue, B.: Cloud detection algorithm comparison and validation for operational Landsat data products, Remote Sensing of Environment, 194, 379–390. https://doi.org/10.1016/j.rse.2017.03.026, 2017.

Line 191 "We excluded estimates derived from tussock measurements because burn depth estimates from tussocks correlated negatively with remotely sensed fire severity.": Is this a part of the analysis that isn't included in the manuscript? More information needed.

RESPONSE: We had misplaced the supplemental figure reference that should have been at the end of this sentence at the end of the prior sentence. Figure A4 [original submission], which will be properly referenced, contains the analysis we conducted to show that "tussock measurements correlated negatively with remotely sensed fire severity." While we did not further analyze explicitly what drove this trend, due to resource and time limitations in the field, we will add hypotheses and an imperative for future work on the subject in this paragraph.

Line 199 "low severity burn": Unburned islands almost certainly exist within the burn scar that the authors focused on. Since there is no "unburned area" class, the "low severity" class most likely includes unburned areas, which is why this class should be called something like "unburned/low severity".

REPONSE: Thank you for pointing out this additional uncertainty in our analysis. Here, we will note the potential inclusion of unburned islands within low severity areas. Throughout the manuscript we will rename the low severity burn class to low severity/unburned. In the discussion of caveats of our study and in the results, we will mention this as a potential source of uncertainty.

Line 239 "similar to the framework employed in Randerson et al. (2006) and Huang et al. (2016).": Even though you provided the references for the method you used in this study, a basic description of the said method should be provided in this manuscript since the audience, including the reviewer, should be able to understand your method without going to the source materials. Also, when directly apply the methods described in previous studies, you need to briefly explain the settings of those studies so that you can show the audience that the methods are actually applicable.

RESPONSE: We agree that our original wording implies we are using methods from these studies, but we do not detail said methods or their applicability. Our intention with these references was to show that the general framework of calculating radiative forcings based on combustion metrics is standard practice, rather than that our methods were taken explicitly from either of these publications. Where our methods adopt specific workflows and equations is detailed throughout the rest of our methods section. We will reword this sentence with these citations to say that our approach is standard practice, but not yet applied to tundra systems in our specific manner.

Line 240 "these equations": Similar to my previous comment, here "these equations" sounds really strange since no equations were actually given in this paper.

RESPONSE: Similar to our previous response, we agree that "these equations" is ambiguous. We will rewrite the sentence to explain that our model was driven by the combustion metrics we calculate. We will also switch the sentence with the one prior for clarity. The final text will read:

> "The radiative forcings model was driven using the average amount of organic matter lost across fire-wide burn severity classes and vegetative reference points. Computing the radiative forcing of gaseous and aerosol emissions has been done for boreal fires (e.g., Huang et al. 2016, O'Halloran et al., 2012, Randerson et al., 2006), but has yet to be applied to tundra systems."

Huang, S., Liu, H., Dahal, D., Jin, S., Li, S. and Liu, S.: Spatial variations in immediate greenhouse gases and aerosol emissions and resulting radiative forcing from wildfires in interior Alaska, Theoretical and applied climatology, 123, 581-592, https://doi.org/10.1007/s00704-015-1379-0, 2016.

O'Halloran, T. L., Law, B. E., Goulden, M. L., Wang, Z., Barr, J. G., Schaaf, C., Brown, M., Fuentes, J. D., Göckede, M. and Black, A.: Radiative forcing of natural forest disturbances, Global Change Biol., 18, 555-565, https://doi.org/10.1111/j.1365-2486.2011.02577.x, 2012.

Randerson, J. T., Liu, H., Flanner, M. G., Chambers, S. D., Jin, Y., Hess, P. G., Pfister, G., Mack, M. C., Treseder, K. K. and Welp, L. R.: The impact of boreal forest fire on climate warming, Science, 314, 1130-1132, https://doi.org/10.1126/science.1132075, 2006.

Line 307-310: Throughout this paper, there are many paragraphs like this that consist of a few sentences. This not a good scientific writing practice and these paragraphs should be rewritten to be merged into bigger paragraphs.

RESPONSE: Here and throughout the paper, we will combine these small paragraphs to make larger ones.

Line 352 "Both models were corrected for spatial autocorrelation between transect locations.": how?

RESPONSE: We will specify that it was in choosing the model with the lowest AIC score across five correlation structures, described in the following sentence, that we corrected for spatial autocorrelation.

Line 410-418: As i pointed out previously, your low severity class includes unburned areas, as a result, you likely have overestimated the emissions here.

RESPONSE: Here and in the discussion, we will note our possible overestimation of emissions, because the low severity areas likely contain patches of unburned tundra.

Line 458 "here we ... frequent tundra fire regimes": this statement needs to be modified significantly. what you can confidently say is that the particular wildfire that you visited emitted a lot of carbon and has a warming effect. However, with the field data that you collected, you can't make a statement indicating this is the case for all tundra wildfires. In fact, this is very

likely not the case for many, if not most tundra wildfires, since tundra wildfires can be quite low in severity.

RESPONSE: We modify this first discussion sentence to focus on our fire season singularly rather than all tundra fires. Additionally, rather than suggesting that all tundra fires represent an under-studied source of greenhouse gas emissions, we will state that our findings suggest increasingly frequent tundra fire regimes, which may include high severity fires, are an under-studied source of global greenhouse gas emissions. The final text will read:

"Here we describe a tundra wildfire season that combusted significant amounts of carbon and had a positive warming effect on the atmosphere due to its gaseous and aerosol emissions over an 80-year time horizon. Our findings suggest that increasingly frequent tundra fire regimes are an under-studied source of global GHG emissions."

Line 463: I don't like how this statement is worded because it incites misinterpretation. You should just say that the emission per unit area is similar between this fire and ARF. That is fair. Additionally, you are comparing this fire with ARF (which is also fair), but you didn't give any other comparison between the two (such as size, species composition, burn severity). These contexts are important for readers to understand the scientific importance of your findings.

RESPONSE: We agree the wording of this sentence is misleading. We will separate it into two sentences to describe the Anaktuvuk River fire separately and added context for that fire, including that it was twice the size, of mostly moderate to high severity, and in tussock tundra. The sentences will read:

"In total, we estimate that about 0.911 Tg of carbon was released from 54,154 hectares of burned area in the YKD…[other inserted material for revisions]... Total carbon loss was driven by a similar per unit area carbon emission rate as the 2007 Anaktuvuk River fire (Mack et al., 2011), which was roughly twice the size of the one reported here and burned predominately moderate to high severity in a tussock tundra ecosystem (Jones et al., 2009)."

Jones, B. M., Kolden, C. A., Jandt, R., Abatzoglou, J. T., Urban, F. and Arp, C. D.: Fire behavior, weather, and burn severity of the 2007 Anaktuvuk River tundra fire, North Slope, Alaska, Arctic, Antarctic, and Alpine Research, 41, 309-316, https://doi.org/10.1657/1938-4246-41.3.309, 2009.

Mack, M. C., Bret-Harte, M. S., Hollingsworth, T. N., Jandt, R. R., Schuur, E. A., Shaver, G. R. and Verbyla, D. L.: Carbon loss from an unprecedented Arctic tundra wildfire, Nature, 475, 489-492, https://doi.org/10.1038/nature10283, 2011.

Line 473"Carbon loss per area from tundra wildfires are within the range of total above- and belowground carbon loss from boreal wildfires, approximately 0.5 to 4 kg m-2 (Walker et al., 2020a; Walker et al., 2018b; Rogers et al., 2014)": This is another instance of overgeneralization.

While this fire and ARF may have carbon emission per unit area that is similar to that of boreal wildfires, you can't say so for all tundra wildfires.

RESPONSE: We will tone back the decisiveness of our conclusion in this sentence by specifying "loss per area from [these] tundra wildfires." We will also adjust the concluding sentence of that paragraph from "While carbon loss per area of tundra wildfire is similar to boreal wildfires…" to say "Although data from the Anaktuvuk River and YKD fires shows carbon loss per unit area can be similar between boreal and tundra wildfires…"

As mentioned in our general response, we will add a comparison of per unit carbon emissions between the fire season measured in our study and remotely sensed combustion of tundra and boreal systems in Alaska with text and an appendix figure. This comparison shows that with respect to remotely sensed combustion measurements, our *in situ* per unit area measurements are on the low end but comparable. We will note that while more representative field measurements are needed in tundra ecosystems to clarify the relationship between tundra and boreal emission rates, the data from our fire season suggests that tundra wildfires should be studied more intensively from a climate perspective.

---

## Author Response (AR1)

**Referee #1**

Wildfires play an important role in the carbon cycle and Earth's energy balance. Despite increasing incidences of tundra wildfires, very few studies on tundra-wildfire carbon emissions exist, and the radiative forcing associated with these emissions is not available. The manuscript of Moubarak et al. attempts to estimate carbon emissions and the resultant radiative forcings from tundra wildfires in the Yukon-Kuskokwim River Delta (YKD), Alaska. While it shows potential scientific significance to the climate community, I find it hard to read overall. It is unclear to me how the authors estimate the carbon emissions and radiative forcing, and the current estimate of radiative forcing is not convincing to me either.

RESPONSE: Thank you for your helpful feedback and the chance to clarify our methodology and importance of our results.  Below we have responded to your general and line-specific comments individually.

General comments

1. The manuscript fails to show the total global amounts of carbon emissions from tundra wildfires and how significant they are, for example, compared to those from the boreal forest on a global basis. While the current estimate of emissions from the YKD area within Alaska is essential, it is not clear whether the global emissions are still significant from the global perspective, given that the Tundra ecosystem covers only a small fraction of the earth.

RESPONSE: Given that tundra fires burn less area than boreal fires, with potentially comparable per unit area emissions, they contribute less to global carbon emissions than boreal fires.  We included a statement making this comparison in the discussion, where we cite the relative burned area for tundra and boreal fires in Alaska between the years 2001 and 2018 (Lines 596-599**)**.  During this time frame, there was a yearly average of 0.5 Mha of boreal burned area compared to 0.017 Mha of tundra (Scholten et al., 2021).  Despite the smaller burned area, tundra fires play an important role in ecosystem function and global climate because they can contribute to permafrost thaw, lack post-fire cooling effects from albedo changes seen in boreal fires, and are increasing in frequency.  These factors heighten their importance in climate change. We highlighted these effects with our statement about global emission totals (Lines 599-600).

Scholten, R.C., S. Veraverbeke, R. Jandt, E.A. Miller, and B.M. Rogers.: ABoVE: Ignitions, Burned Area, and Emissions of Fires in AK, YT, and NWT, 2001-2018, ORNL DAAC, Oak Ridge, Tennessee, USA [data set], https://doi.org/10.3334/ORNLDAAC/1812, 2021.

2. The radiative forcing estimation in the current format looks faulty. First, the radiative forcing is given as a global average. However, the emissions and concentrations used here are the regional mean values. In addition, how do the authors translate the emissions to the concentrations of the gas/aerosol remaining in the atmosphere?

RESPONSE: Calculating global averages is standard practice for radiative forcing modeling of regional wildfires (e.g., Huang et al., 2016, O'Halloran et al., 2012, and Randerson et al., 2006). We more explicitly state our method's precedent with these citations in the methods section, '2.7: Radiative forcings model' (Lines 262-263). Our final estimate for radiative forcing from this tundra wildfire season was bounded by estimates of the gaseous radiative forcing from boreal wildfires (Huang et al., 2016, Randerson et al., 2006). Thus, our calculation is within reason, especially given the similar levels of per unit area combustion we show between the fire season we measure and boreal fires (Lines 511-524). We more explicitly contextualize our results with this comparison in the discussion by stating that the radiative forcing we calculated for these tundra wildfires is within the range of estimates for the gaseous radiative forcing of boreal fires (Lines 578-580).

We translated from emissions to atmospheric concentration of gaseous species remaining by converting from a mass emitted (calculated using emissions factors) to a volume as a molar fraction of the atmosphere. This step was crucial in converting from a regional emission to a global value, and we make it more clear in our methods in response to the reviewer's line-specific suggestions below (Lines 293-295, 318-320, 331). Such a conversion was possible and is standard because greenhouse gasses are generally long-lived and globally-mixed, and global averages allow for common comparison.

Huang, S., Liu, H., Dahal, D., Jin, S., Li, S. and Liu, S.: Spatial variations in immediate greenhouse gases and aerosol emissions and resulting radiative forcing from wildfires in interior Alaska, Theoretical and applied climatology, 123, 581-592, https://doi.org/10.1007/s00704-015-1379-0, 2016.

O'Halloran, T. L., Law, B. E., Goulden, M. L., Wang, Z., Barr, J. G., Schaaf, C., Brown, M., Fuentes, J. D., Göckede, M. and Black, A.: Radiative forcing of natural forest disturbances, Global Change Biol., 18, 555-565, https://doi.org/10.1111/j.1365-2486.2011.02577.x, 2012.

Randerson, J. T., Liu, H., Flanner, M. G., Chambers, S. D., Jin, Y., Hess, P. G., Pfister, G., Mack, M. C., Treseder, K. K. and Welp, L. R.: The impact of boreal forest fire on climate warming, Science, 314, 1130-1132, https://doi.org/10.1126/science.1132075, 2006.

Specific comments

Line 57: Is there anything to do with "more variable precipitation" here? How is it related to the "increasing temperature"?

RESPONSE: By mentioning more variable precipitation in the context of increasing temperature, we meant to highlight the more frequent co-occurrence of hot and dry conditions, which provides context for the climate and environmental drivers of increasing wildfires in the following paragraph. We added the phrase "hence more frequently pairing hot and dry conditions" to make this explicit (Lines 58-59).

Line 164: How do the authors "maximize" physical separation of the unburned sites?

RESPONSE: We removed the word "maximize" since it implies a formal optimization. Instead, we will cite the minimum distance of 1 kilometer between unburned sites (Lines 189-191).

Line 190-191: How do the authors calculate the organic matter combustion and carbon loss? I cannot get any hint from Fig. A4. Can they provide an equation and/or diagram to help readers understand this?

RESPONSE: Referencing Fig. A4 here was a typo, since it was meant to illustrate the trend in tussock burn depth measurements. We moved our reference of the figure to a later sentence in the paragraph where we discuss our exclusion of tussock measurements because their burn depths correlated negatively with fire severity (Line 211).

We also added a schematic to the appendix to illustrate our calculation of organic matter and carbon loss at each burn depth transect (Lines 715-716). This figure is referenced after text that describes how organic matter and carbon loss were calculated (Lines 229-230).

Line 192: Can the authors explain the reasons for the unexpected behaviors of tussock measurements?

REPONSE: Given the limited sample size of tussocks on the landscape and the lack of an explicit experimental design for this specific purpose, we cannot make definitive statements regarding the behavior of tussock measurements. We added the suggestion that tussocks may alter burn dynamics locally, while emphasizing that no conclusions can be drawn from our current dataset (Lines 211-213).

Line 246-248: How about the uncertainty associated with the emission factor?

RESPONSE: Since Akagi et al. (2011) combines measurements of emissions factors for each gaseous species from multiple studies with different designs, such as

laboratory, field, and airborne, uncertainty is introduced into the emissions factor by the measurement technique. Post-emission chemical reactions from the released gaseous species cause the concentrations of specific gasses to vary after emission. As a result, how the gasses are measured might affect the emission factor. We summarized this uncertainty in the methods (Lines 273-277).

Akagi et al. (2011) provides an estimate of natural variation for most emissions factors, which we did not include in our model. We highlighted this in the uncertainty section of our discussion by stating future models might incorporate the natural variation of each emission factor to bound their estimates (Lines 646-647).

Akagi, S. K., Yokelson, R. J., Wiedinmyer, C., Alvarado, M. J., Reid, J. S., Karl, T., Crounse, J. D. and Wennberg, P. O.: Emission factors for open and domestic biomass burning for use in atmospheric models, Atmospheric Chemistry and Physics, 11, 4039-4072, https://doi.org/10.5194/acp-11-4039-2011, 2011.

Line 277-283. How do the authors get the atmospheric concentrations of CH4 and N2O from the fire emissions?

RESPONSE: While we mentioned briefly on line 250 (original submission) in our general description of the radiative forcings model that we converted the mass of the gaseous species emitted to volumes as molar fractions of the atmosphere, we rearranged our text to make this methodology clearer for the gaseous species to which it was applied. First, we removed that original sentence on line 250 (original submission). Then, for $CH_4$ and $N_2O$, we added a sentence in section 2.8 stating that the concentration of both gasses was calculated by converting the initial mass emitted to a volume as a molar fraction of the atmosphere (Lines 294-295). We also originally labeled $C_o$ in equation 3 as an "initial pulse mass of the gas." Here we changed "mass" to "concentration" (Line 293). For $CO_2$, we added a similar sentence describing the conversion of mass to atmospheric concentration in section 2.9 (Lines 318-320). Finally, the same method was also used for CO when calculating the concentration of tropospheric ozone. Therefore, we added the same description for CO in section 2.10 (Line 331).

Line 294. Again, how do the authors estimate the CO2 concentration from the fire emissions?

RESPONSE: See previous response to lines 277-283.

Line 318. Although ozone precursors can affect CH4 formation, how are aerosols related to CH4?

RESPONSE: Our original text on this line was misleading. While aerosols are not explicitly related to CH4 formation, we chose to model them using methods similar to those we used for ozone precursors for the sake of consistency in GWP-derived radiative forcings calculations. We changed the text to reflect this methodological choice by stating: "We derived the radiative forcings for ozone precursors based on $CH_4$ because the radiative forcing of ozone precursors is through their effect on methane over the long-term (Collins et al., 2013). To remain consistent with our methodology for ozone precursors, we derive the radiative forcings for aerosols from $CH_4$ as well" (Lines 344-347).

Collins, W. J., Fry, M. M., Yu, H., Fuglestvedt, J. S., Shindell, D. T. and West, J. J.: Global and regional temperature-change potentials for near-term climate forcers, Atmospheric Chemistry and Physics, 13, 2471-2485, https://doi.org/10.5194/acp-13-2471-2013, 2013.

Line 340. Why is R20 used here? So only accounts for the cumulative radiative forcing at the 20-year time zone?

RESPONSE: Our original text did not fully explain our assumptions for our aerosol calculations. We rearranged the paragraph and add text to clarify (Lines 368-374). R20 is used because the total radiative forcing of aerosols likely happens within the first year of the fire, as aerosols exit the atmosphere on a time scale of weeks. Therefore, the cumulative radiative forcing of aerosols every year after the fire event is constant. We chose to use the GWP of the aerosols at year 20 to calculate the radiative forcing and assumed this value was the same across all years. We changed the text on lines 368-371 to state that:

> "The radiative effect of aerosol emissions happens within a year of the fire event, as fire aerosols are typically removed from the atmosphere via wet and dry deposition within a matter of weeks (Bond et al., 2011, Quinn et al., 2008). We assumed that the cumulative radiative forcing of aerosols at any year after the fire event would be constant and equal to the radiative forcing of aerosols in the year of the fire event."

Bond, T. C., Zarzycki, C., Flanner, M. G. and Koch, D. M.: Quantifying immediate radiative forcing by black carbon and organic matter with the Specific Forcing Pulse, Atmospheric Chemistry and Physics, 11, 1505-1525, https://doi.org/10.5194/acp-11-1505-2011, 2011.

Quinn, P. K., Bates, T. S., Baum, E., Doubleday, N., Fiore, A. M., Flanner, M., Fridlind, A., Garrett, T. J., Koch, D. and Menon, S.: Short-lived pollutants in the Arctic: their

climate impact and possible mitigation strategies, Atmospheric Chemistry and Physics, 8, 1723-1735, https://doi.org/10.5194/acp-8-1723-2008, 2008.

Line 350: what is the "gls" function?

RESPONSE: To make clear the "gls" function is defined by the nlme package, we rearranged the sentence to say "we used the nlme function "gls" to fit a linear model". We also clarified that the model was fit using the generalized least squares method (Lines 380-381).

Line 359: what is the "lme" function?

RESPONSE: Similarly, to emphasize the "lme" function came from the nlme package, we added the phrase "defined in the nlme package". We also clarified that the model is fit using the restricted maximum likelihood method (Lines 389-390).

Line 396: what is "SEM"?

RESPONSE: SEM stands for standard error of the mean, and the text was changed to reflect this definition (Line 426).

**Referee #2**

The following comments pertain to the manuscript "Yukon-Kuskokwim River Delta 2015 fire burn depth measurements and unburned soil and vegetation organic matter and carbon content collected in 2019". Overall, I think the findings presented in this manuscript are of high scientific importance and I applaud the authors for conducting this work. Having said that, I have a few major issues with this manuscript. The first one concerns the language of this manuscript. If my understanding of this manuscript is correct, the findings presented here stemmed from field data collected in a single burn scar in YKD. While these findings are important in their own rights, the authors (mostly in the results and discussion sessions) made a lot of statements that are obviously overly strong. Tundra wildfires, while being overall understudied, vary substantially based on climate, terrain, species composition, etc. As a result, while this study shows that the YKD fire that it looked into had a similar emission rate to the Anaktuvuk River Fire (ARF), they still represent two cases out of many more tundra fires, many of which are much less severe than the ARF. Therefore, the authors should be very careful when they make statements concerning "tundra fires". You can indicate that the YKD fire and the ARF likely represent the upper level of the impacts of tundra fires (in terms of carbon emission and radiative forcing), but you can't just hint that these are representative situations.

My second major issue with this manuscript is the fact that while it has a substantial discussion session, it fails to discuss the sources of uncertainty introduced by the methods that were used. For example, this study used dNBR as the indicator for burn severity, however, there have been studies that show that dNBR may not be a very good burn severity indicator in the tundra (Loboda et al 2013) and that there may be better indicators (Chen et al 2020). Another major potential source of uncertainty stems from the fact that the emission factors the authors used were from boreal fires, which as the authors state in the method section may lead to an overestimation. However, this overestimation wasn't further discussed in the discussion.

Additionally, in terms of dNBR, there have been a lot of studies showing the caveats associated with dNBR when it is used in high northern latitude settings (eg, Chen et al 2020). The author failed to take these into account completely. Additionally, the authors never showed the audience how the YKD fire that they picked compared with other tundra fires. This is a piece of critical information that is needed for a wider audience to understand the scientific importance of your work.

Loboda, T. V., French, N. H., Hight-Harf, C., Jenkins, L., & Miller, M. E. (2013). Mapping fire extent and burn severity in Alaskan tussock tundra: An analysis of the spectral response of tundra vegetation to wildland fire. Remote Sensing of Environment, 134, 194-209.

Chen, Yaping, Mark Jason Lara, and Feng Sheng Hu. "A robust visible near-infrared index for fire severity mapping in Arctic tundra ecosystems." ISPRS Journal of Photogrammetry and Remote Sensing 159 (2020): 101-113.

Chen, D., Loboda, T. V., & Hall, J. V. (2020). A systematic evaluation of influence of image selection process on remote sensing-based burn severity indices in North American boreal forest and tundra ecosystems. ISPRS Journal of Photogrammetry and Remote Sensing, 159, 63-77.

RESPONSE: Thank you for your helpful feedback, as it allows us to more accurately represent the scope of our findings and their importance to the scientific community. While our initial text generalized our results to all tundra fires, we both scaled back the language of our results and provided context for the fire season we measured compared to other fires in Alaska.  First, we clarified that our results represent combustion from a single tundra fire season in Alaska and may not be representative for all tundra wildfires.  Instead of generalizing our results to all tundra fires, we stated that on a per unit area basis, the fire season measured herein, the ARF fire, and boreal fires have relatively similar levels of combustion, which suggests emissions rates per unit burned area from tundra fires can be comparable to boreal fires.  These changes will be

made in the discussion and in response to your line-specific comments below (Lines 492-495, 512, 522-523)

In addition, we compared per unit area combustion from our study with all fires between 2001 and 2018 in boreal and tundra systems in Alaska, estimated via remote sensing and geospatial machine learning techniques, by adding a density plot to the appendix (Lines 724-727).  While field measurements of tundra combustion are currently lacking, but necessary to validate remotely sensed combustion products, our field measurements fell within the range of remotely sensed combustion in tundra and boreal systems.  We made note of this trend in the text by stating that:

> "Compared to remote sensing-derived estimates of combustion, our per unit area *in situ* measurements are on the low end but comparable to large-scale means across Alaska between 2001 and 2018 (Fig. A4 [Lines 724-727]).  However, more representative field measurements in tundra fires are needed to validate remotely sensed combustion measurements, as the currently available gridded products (Potter et al. 2022; Scholten et al. 2021; Veraverbeke et al. 2017a; Veraverbeke et al., 2015a) are driven entirely by measurements in boreal forests" (Lines 517-522).

In addition to adding this context in the discussion, we added context for our fire season's size and dNBR values in the methods (Lines 130-137), results (Lines 440-442), and discussion (Lines 497-501) sections.  The 2015 YKD fire season burned roughly three times the annual average tundra burned area in Alaska between the years 2001 and 2018, but it accounted for only a little more than half of the Alaskan tundra area burned in 2015 alone.  Within the YKD, the 2015 fire season burned more area than any other year between 2001 and 2018.  Most years in that time frame burned less than 10,000 hectares.  In comparison, the 2015 YKD fire season burned 54,154 hectares.  This context was added in the first paragraph of the discussion (Lines 497-501).  The dNBR values for the fire season we studied were intermediate compared to dNBR values for tundra and on the low side but within the range of values for boreal fires in Alaska generally (Lines 440-442).  These contextual changes clarify that while definitive conclusions about how our fire season translates to other tundra fires cannot be made presently, the results of our study highlight the need for more attention to tundra wildfires from a climate perspective.  We elaborate more on these contextual changes in response to your line-specific comment below as well.

We expanded the discussion section describing sources of uncertainty introduced by our methodology to add more description of the sources you mention here.  First, we described that despite dNBR being the most widely used metric to

assess fire severity, its accuracy may be diminished in high latitude tundra systems by environmental and methodological factors. We appreciate the studies you suggested, and we cited them in our manuscript. In the text, we noted that other metrics have shown promise in tundra systems, but how these results translate to other tundra systems remains to be explored (Lines 625-630). Second, throughout the manuscript we noted that our low severity fire class likely includes unburned patches of tundra, leading us to potentially overestimate combustion in low severity areas (Lines 178-180, 448-449, 629-630, low severity class renamed - e.g., line 175). Finally, we reiterated in the discussion that since emissions factors for tundra fires are not available, we used boreal emissions factors, potentially leading us to overestimate gaseous emissions from woody vegetation (Lines 643-646). As noted in the discussion, our model was parameterized with data from various sources and is publicly available via GitHub. Since data on tundra wildfires is lacking to date, our parameterization could only approximate tundra fires. However, as these parameters are updated for tundra systems in future literature, our model can be updated. In addition to this general response, we address your detailed comments individually below.

References for general comment:

Potter, S., Veraverbeke, S., Walker, X. J., Mack, M. C., Goetz, S. J., Baltzer, J. L., Dieleman, C., French, N. H. F., Kane, E. S., Turetsky, M. R., Wiggins, E. B., and Rogers, B. M.: ABoVE: Burned Area, Depth, and Combustion for Alaska and Canada, 2001-2019, ORNL DAAC, Oak Ridge, Tennessee, USA [data set], https://doi.org/10.3334/ORNLDAAC/2063, 2022.

Scholten, R.C., S. Veraverbeke, R. Jandt, E.A. Miller, and B.M. Rogers.: ABoVE: Ignitions, Burned Area, and Emissions of Fires in AK, YT, and NWT, 2001-2018, ORNL DAAC, Oak Ridge, Tennessee, USA [data set], https://doi.org/10.3334/ORNLDAAC/1812, 2021.

Veraverbeke, S., Rogers, B. M., and Randerson, J. T.: CARVE: Alaskan Fire Emissions Database (AKFED), 2001-2013, ORNL DAAC, Oak Ridge, Tennessee, USA [data set], https://doi.org/10.3334/ORNLDAAC/1282, 2015a.

Veraverbeke, S., Rogers, B. M., Goulden, M. L., Jandt, R., Miller, C. E., Wiggins, E. B., and Randerson, J. T.: ABoVE: Ignitions, burned area and emissions of fires in AK, YT, and NWT, 2001-2015, ORNL DAAC, Oak Ridge, Tennessee, USA [data set], https://doi.org/10.3334/ORNLDAAC/1341, 2017a.

Here are my more detailed comments.

Line 38 "AK": Spell out Alaska.

RESPONSE: Changed (Line 38).

Line 64 "North slope": North Slope

RESPONSE: Changed (Line 65).

Line 97: More information about how this particular fire compares with other tundra fires (both from YKD and other tundra regions in Alaska) is needed (in terms of size, overall dNBR, etc).

RESPONSE: We added a few pieces of information to put the YKD 2015 tundra fire season in context. We added this information throughout the methods, results, and discussion where it is appropriate. First, as stated in response to your general comment, we made comparisons based on burned area in the first paragraph of the discussion (Lines 497-501). The 2015 YKD fire season burned roughly three times the average annual burned area in Alaskan tundra between the years 2001 and 2018, but it accounted for only a little more than half of the tundra area burned in Alaska in the year 2015. The 2015 fire season had the largest burned area within the YKD compared to the years 2001 to 2018. Most years in that time frame had burned areas less than 10,000 hectares. In comparison, the 2015 YKD fire season burned 54,154 hectares.

Also, we compared the dNBR values from our study to dNBR values for all fires in Alaskan tundra and boreal systems. Differenced NBR for the fire season in this study was intermediate to dNBR for all tundra fires in Alaska from 1989 to 2019 and on the low side but within the range of dNBR for Alaskan boreal fires during this same timeframe. We added our means of obtaining this information to the methods (Lines 130-137) and the data is cited in the results (Lines 440-442) and as a density plot in the appendix (Lines 721-723). As mentioned in our general comment, we compared the per unit area combustion from our focal fires to other remotely sensed per unit area combustion measurements for tundra fires in Alaska (Lines 517-522, 725-727).

Line 105: Where did you get the burn scars? This needs to be specified.

RESPONSE: The burn scars were found using the Alaska Large Fire Database (Kasischke et al., 2002). We now mention this in the current section 2.1 (Lines 104-105, 110) and the methodology section 2.2 that describes fire severity (Line 121).

Kasischke, E. S., Williams, D. and Barry, D.: Analysis of the patterns of large fires in the boreal forest region of Alaska. International Journal of Wildland Fire, 11, 131–144, https://doi.org/10.1071/wf02023, 2002.

Line 181 dNBR's full name has already been given previously.

RESPONSE: We replaced the full name with the acronym (Line 122).

Line 181 "Key and Benson 2006": This reference should be provided when dNBR was first mentioned.

RESPONSE: We added this citation where dNBR is first mentioned in the introduction (Line 93).

Line 183: CFMASK algorithm: this needs a citation.

RESPONSE: We added the citation Foga et al. (2017) (Line 124).

Foga, S., Scaramuzza, P. L., Guo, S., Zhu, Z., Dilley, R. D., Beckmann, T., Schmidt, G. L., Dwyer, J. L., Joseph Hughes, M., and Laue, B.: Cloud detection algorithm comparison and validation for operational Landsat data products, Remote Sensing of Environment, 194, 379–390. https://doi.org/10.1016/j.rse.2017.03.026, 2017.

Line 191 "We excluded estimates derived from tussock measurements because burn depth estimates from tussocks correlated negatively with remotely sensed fire severity.": Is this a part of the analysis that isn't included in the manuscript? More information needed.

RESPONSE: We had misplaced the supplemental figure reference that should have been at the end of this sentence at the end of the prior sentence. Figure A4, which is now properly referenced (Lines 208-211), contains the analysis we conducted to show that "tussock measurements correlated negatively with remotely sensed fire severity." While we did not further analyze explicitly what drove this trend, due to resource and time limitations in the field, we added hypotheses and an imperative for future work on the subject in this paragraph (Lines 211-213).

Line 199 "low severity burn": Unburned islands almost certainly exist within the burn scar that the authors focused on. Since there is no "unburned area" class, the "low severity" class most likely includes unburned areas, which is why this class should be called something like "unburned/low severity".

REPONSE: Thank you for pointing out this additional uncertainty in our analysis. Here, we now note the potential inclusion of unburned islands within low severity areas (Line 219). We also note this uncertainty where we originate the burn severity class definitions (Lines 178-180). Throughout the manuscript we renamed the low severity burn class to low severity/unburned (e.g., lines 215, 217). In the discussion of caveats of our study (Lines 629-630) and in the results (Lines 448-449), we now mention this as a potential source of uncertainty.

Line 239 "similar to the framework employed in Randerson et al. (2006) and Huang et al. (2016).": Even though you provided the references for the method you used in this

study, a basic description of the said method should be provided in this manuscript since the audience, including the reviewer, should be able to understand your method without going to the source materials. Also, when directly apply the methods described in previous studies, you need to briefly explain the settings of those studies so that you can show the audience that the methods are actually applicable.

RESPONSE: We agree that our original wording implies we are using methods from these studies, but we do not detail said methods or their applicability. Our intention with these references was to show that the general framework of calculating radiative forcings based on combustion metrics is standard practice, rather than that our methods were taken explicitly from either of these publications. Where our methods adopt specific workflows and equations is detailed throughout the rest of our methods section. We reworded this sentence with these citations to say that our approach is standard practice, but not yet applied to tundra systems in our specific manner (Lines 262-263).

Line 240 "these equations": Similar to my previous comment, here "these equations" sounds really strange since no equations were actually given in this paper.

RESPONSE: Similar to our previous response, we agree that "these equations" is ambiguous. We rewrote the sentence to explain that our model was driven by the combustion metrics we calculate. We also switched the sentence with the one prior for clarity. The final text reads:

> "The radiative forcings model was driven using the average amount of organic matter lost across fire-wide burn severity classes and vegetative reference points. Computing the radiative forcing of gaseous and aerosol emissions has been done for boreal fires (e.g., Huang et al. 2016, O'Halloran et al., 2012, Randerson et al., 2006), but has yet to be applied to tundra systems" (Lines 260-263)**.**

Huang, S., Liu, H., Dahal, D., Jin, S., Li, S. and Liu, S.: Spatial variations in immediate greenhouse gases and aerosol emissions and resulting radiative forcing from wildfires in interior Alaska, Theoretical and applied climatology, 123, 581-592, https://doi.org/10.1007/s00704-015-1379-0, 2016.

O'Halloran, T. L., Law, B. E., Goulden, M. L., Wang, Z., Barr, J. G., Schaaf, C., Brown, M., Fuentes, J. D., Göckede, M. and Black, A.: Radiative forcing of natural forest disturbances, Global Change Biol., 18, 555-565, https://doi.org/10.1111/j.1365-2486.2011.02577.x, 2012.

Randerson, J. T., Liu, H., Flanner, M. G., Chambers, S. D., Jin, Y., Hess, P. G., Pfister, G., Mack, M. C., Treseder, K. K. and Welp, L. R.: The impact of boreal forest fire on climate warming, Science, 314, 1130-1132, https://doi.org/10.1126/science.1132075, 2006.

Line 307-310: Throughout this paper, there are many paragraphs like this that consist of a few sentences. This not a good scientific writing practice and these paragraphs should be rewritten to be merged into bigger paragraphs.

RESPONSE: Here and throughout the paper, we combined these small paragraphs to make larger ones (Lines 127, 227, 304, 307, 334, 388, 392, 414)

Line 352 "Both models were corrected for spatial autocorrelation between transect locations.": how?

RESPONSE: We specified that it was in choosing the model with the lowest AIC score across five correlation structures, described in the following sentence, that we corrected for spatial autocorrelation (Lines 382-385).

Line 410-418: As i pointed out previously, your low severity class includes unburned areas, as a result, you likely have overestimated the emissions here.

RESPONSE: Here and in the discussion, we now make note of our possible overestimation of emissions, because the low severity areas likely contain patches of unburned tundra (Lines 448-449, 629-630).

Line 458 "here we … frequent tundra fire regimes": this statement needs to be modified significantly. what you can confidently say is that the particular wildfire that you visited emitted a lot of carbon and has a warming effect. However, with the field data that you collected, you can't make a statement indicating this is the case for all tundra wildfires. In fact, this is very likely not the case for many, if not most tundra wildfires, since tundra wildfires can be quite low in severity.

RESPONSE: We modified this first discussion sentence to focus on our fire season singularly rather than all tundra fires. Additionally, rather than suggesting that all tundra fires represent an under-studied source of greenhouse gas emissions, we now state that our findings suggest increasingly frequent tundra fire regimes, which may include high severity fires, are an under-studied source of global greenhouse gas emissions. The final text reads:

> "Here we describe a tundra wildfire season that combusted significant amounts of carbon and had a positive warming effect on the atmosphere due to its gaseous and aerosol emissions over an 80-year time horizon. Our findings suggest that increasingly frequent tundra fire regimes are an under-studied source of global GHG emissions" (Lines 492-495).

Line 463: I don't like how this statement is worded because it incites misinterpretation. You should just say that the emission per unit area is similar between this fire and ARF. That is fair. Additionally, you are comparing this fire with ARF (which is also fair), but you didn't give any other comparison between the two (such as size, species composition, burn severity). These contexts are important for readers to understand the scientific importance of your findings.

RESPONSE: We agree the wording of this sentence is misleading.  We separated it into two sentences to describe the Anaktuvuk River fire separately and added context for that fire, including that it was twice the size, of mostly moderate to high severity, and in tussock tundra.  The sentences now read:

> "In total, we estimate that about 0.911 Tg of carbon was released from 54,154 hectares of burned area in the YKD…[other inserted material for revisions]... Total carbon loss was driven by a similar per unit area carbon emission rate as the 2007 Anaktuvuk River fire (Mack et al., 2011), which was roughly twice the size of the one reported here and burned predominately moderate to high severity in a tussock tundra ecosystem (Jones et al., 2009)" (Lines 496-497, 501-504).

Jones, B. M., Kolden, C. A., Jandt, R., Abatzoglou, J. T., Urban, F. and Arp, C. D.: Fire behavior, weather, and burn severity of the 2007 Anaktuvuk River tundra fire, North Slope, Alaska, Arctic, Antarctic, and Alpine Research, 41, 309-316, https://doi.org/10.1657/1938-4246-41.3.309, 2009.

Mack, M. C., Bret-Harte, M. S., Hollingsworth, T. N., Jandt, R. R., Schuur, E. A., Shaver, G. R. and Verbyla, D. L.: Carbon loss from an unprecedented Arctic tundra wildfire, Nature, 475, 489-492, https://doi.org/10.1038/nature10283, 2011.

Line 473"Carbon loss per area from tundra wildfires are within the range of total above- and belowground carbon loss from boreal wildfires, approximately 0.5 to 4 kg m-2 (Walker et al., 2020a; Walker et al., 2018b; Rogers et al., 2014)": This is another instance of overgeneralization. While this fire and ARF may have carbon emission per unit area that is similar to that of boreal wildfires, you can't say so for all tundra wildfires.

RESPONSE: We toned back the decisiveness of our conclusion in this sentence by specifying "loss per area from [these] tundra wildfires" (Line 512).  We also adjusted the concluding sentence of that paragraph from "While carbon loss per area of tundra wildfire is similar to boreal wildfires…" to say, "Although data from the Anaktuvuk River and YKD fires shows carbon loss per unit area can be similar between boreal and tundra wildfires…" (Lines 522-523).

As mentioned in our general response, we added a comparison of per unit carbon emissions between the fire season measured in our study and remotely sensed combustion of tundra and boreal systems in Alaska with text and an appendix figure. This comparison shows that with respect to remotely sensed combustion measurements, our *in situ* per unit area measurements are on the low end but comparable.  We now note that while more representative field measurements are needed in tundra ecosystems to clarify the relationship between tundra and boreal emission rates, the data from our fire season suggests that tundra wildfires should be studied more intensively from a climate perspective (Lines 517-524).

---

## Author Response (AR2)

**Reviewer 1**

Thanks to the authors for their clarification and revisions. I have two more comments/questions below.

Thank you for your additional comments and questions on our manuscript. We appreciate the chance to clarify uncertainties in our radiative forcing calculations, put the importance of tundra wildfires into a global perspective, and propose directions for future research. We responded to your specific comments individually below and made changes in blue to the manuscript. Red text denotes changes from the first round of review.

1. As the burned areas of tundra fires are much smaller than those of boreal forest fires, is the global impact of tundra fires supposed to be limited? Can you compare the radiative forcing of tundra-fire emissions to that due to boreal forest fires?

RESPONSE: While to our knowledge there has only been limited study quantifying the net global radiative forcing of boreal fires (Oris et al., 2014), these questions raise an important point regarding the net effect of wildfires, including both gaseous emissions and ecosystem changes. On lines 578-580 we show that our estimate of the radiative forcing of gaseous emissions from tundra wildfires is within the range of previously estimated radiative forcings for the gaseous emissions from boreal wildfires. However, boreal wildfires have a negative radiative forcing from post-fire changes in surface albedo that has the potential to partially or completely counterbalance the warming effect of gaseous emissions. Without overstory vegetation, tundra ecosystems lack this cooling post-fire albedo effect to balance the warming effect of their gaseous emissions and other post-fire ecosystem changes. These competing dynamics are discussed between lines 580 and 596. We add text on lines 600-602 to suggest that future work synthesizes the net radiative forcing of boreal and tundra wildfires across their global domains. It could be the case that despite the smaller size and lower magnitude of gaseous emissions, tundra wildfires have a comparable or greater global impact compared to boreal fires per unit area when accounting for their additional warming post-fire ecosystem changes and lack of a cooling albedo effect (lines 602-605). Furthermore, even though tundra wildfires currently burn less area than boreal wildfires in both Alaskan (lines 596-597) and pan-Arctic (lines 597-599) regions, the projected increase in tundra fire frequency (e.g., lines 531-534) may also contribute to a higher net radiative forcing from tundra than boreal wildfires in the future (lines 602-605).

Reference:
Oris, F., Asselin, H., Ali, A. A., Finsinger, W., and Bergeron, Y.: Effect of increased fire activity on global warming in the boreal forest, Environmental Reviews, 22, 206–219, https://doi.org/10.1139/er-2013-0062, 2014.

2. Emission factors are crucial for the radiative forcing estimates in this paper. However, the authors did not measure the emission factors of tundra emissions. Instead, they used the emission factors of boreal forest fires. While the greenhouse gases are long-lived and well-mixed in the atmosphere, the aerosols are short-lived and quite heterogeneous. So the authors should at least discuss the limitation and the uncertainty of the radiative forcing calculations.

RESPONSE: We agree that our use of boreal emissions factors and estimation of the effect of SCLFs inject uncertainty into our radiative forcing calculations.  We made textual changes in the paragraph of our discussion section that enumerates uncertainties in our radiative forcing model to reflect these caveats. On lines 651-652 we added text describing how the use of boreal emissions factors may misrepresent the specific mass of gaseous emissions from tundra biomass burning.  We also suggest that future studies directly measure emissions factors from tundra biomass burning (lines 654-655).  On lines 655-658, we describe how our radiative forcing calculations are done with and without taking SCLFs into account, because their effect when emitted from the Arctic remains uncertain in current literature.  We suggest that future research examine how emission location, abbreviated atmospheric lifetime, and regional atmospheric patterns govern the behavior of SCLFs to constrain estimates of radiative forcing.